# A Unified Diversity Measure for Multiagent Reinforcement Learning

**Zongkai Liu**
School of Computer Science and Engineering
Sun Yat-sen University, Guangzhou, China
liuzk@mail2.sysu.edu.cn

**Chao Yu**[*]
School of Computer Science and Engineering
Sun Yat-sen University, Guangzhou, China
yuchao3@mail.sysu.edu.cn

**Yaodong Yang**
Institute for AI, Peking University
Beijing Institute for General AI, Beijing
yaodong.yang@pku.edu.cn

**Peng Sun**
ByteDance, Shenzhen, China
pengsun000@gmail.com

**Zifan Wu**
School of Computer Science and Engineering
Sun Yat-sen University, Guangzhou, China
wuzf5@mail2.sysu.edu.cn

## Abstract

Promoting behavioural diversity is of critical importance in multi-agent reinforcement learning, since it helps the agent population maintain robust performance when encountering unfamiliar opponents at test time, or, when the game is highly non-transitive in the strategy space (e.g., Rock-Paper-Scissor). While a myriad of diversity metrics have been proposed, there are no widely accepted or unified definitions in the literature, making the consequent diversity-aware learning algorithms difficult to evaluate and the insights elusive. In this work, we propose a novel metric called the *Unified Diversity Measure* (UDM) that offers a unified view for existing diversity metrics. Based on UDM, we design the *UDM-Fictitious Play* (UDM-FP) and *UDM-Policy Space Response Oracle* (UDM-PSRO) algorithms as efficient solvers for normal-form games and open-ended games. In theory, we prove that UDM-based methods can enlarge the *gamescape* by increasing the response capacity of the strategy pool, and have convergence guarantee to two-player Nash equilibrium. We validate our algorithms on games that show strong non-transitivity, and empirical results show that our algorithms achieve better performances than strong PSRO baselines in terms of the *exploitability* and *population effectivity*.

## 1 Introduction

Diversity is a widely studied topic in machine learning, including autonomous driving [52], gaming AI [46], recommender systems [6], generative models [13], and latent variable models [51]. Specially, in Multi-Agent Reinforcement Learning (MARL) [4, 20, 55, 50], diversity of agents' strategies is helpful for learning different skills [14], facilitating explorations [42], and discovering sophisticated cooperative policies [27] or an automated agenda towards unexploitable policies in

---

[*]Corresponding authors.

36th Conference on Neural Information Processing Systems (NeurIPS 2022).

competitive games [15, 29]. Promoting diversity of strategies is also an effective method for solving games with non-transitive dynamics [1, 40, 12, 46]. In general, an arbitrary game, of either the normal-form type [5] or the differential type [2], can always be decomposed into a sum of a transitive part and a non-transitive part, where the former represents a transitive rule (i.e., if A beats B, B beats C, then A beats C), while the latter represents a cyclic rule (i.e., the cycles among Rock, Paper and Scissors). In fact, many real-world games demonstrate strong non-transitivity [8], and thus each player must use a diverse pool of winning strategies to achieve low exploitability [53], which has been justified recently by the super-human performance of AIs in sophisticated tasks like StarCraft [47, 43] and DOTA2 [3, 57].

In specific, encouraging the diversity of strategies is critical in achieving high-level learning performance due to the following two main aspects: (1) in the training process, diversity prevents agents from checking the same strategies repeatedly [40], and helps them explore strategy space sufficiently [42]; and (2) at test time, diversity not only prevents agents from being exploited (i.e., a single strategy can always be beaten by another one on non-transitive games [40, 30]), but also helps them learn adaptable strategies and thus maintain robust performances [49, 30].

Due to its great importance, there are a number of studies that investigate the definition of diversity and propose diversity-driven MARL algorithms. The majority of work has followed a heuristic approach [40], such as *evolutionary computation* [16] with a central focus that mimics the natural evolution process. In evolutionary computation, one idea for encouraging diversity is *novelty search* [24] that searches for strategies leading to novel outcomes. *Quality-diversity* (QD) [44] combines novelty search with fitness objective, resulting two representative methods: *Novelty Search with Local Competition* [25] and *MAP-Elites* [7, 37]. Although these methods achieve empirical successes [21, 7], quantifying diversity in evolutionary computation is often hand-crafted and task-dependent [40]. There are also various metrics proposed to model diversity in a more rigorous way, which can be mainly classified into *Behavioral Diversity* (BD) and *Response Diversity* (RD) [30]. A main principle in BD to characterize diversity is to construct metrics over the trajectory or state-action distribution [28, 14, 27, 42]. However, this kind of methods only focus on the behavioral trajectories and ignore the reward attributes of strategies, which may be problematic in scenarios when a slight difference in strategy behaviors will lead to a huge difference in the corresponding reward [30]. Different from BD, RD directly uses the empirical payoff matrix to construct the diversity of strategies [1, 40, 30], and thus it estimates the diversity from the response when encountering distinct opponents.

Despite of the above great efforts, there are still no consistent formal definitions for diversity, making it difficult to evaluate the diverse strategies in MARL. In this paper, we work towards offering a consistent definition for diversity, and propose a novel population-wide diversity measure called the *Unified Diversity Measure* (UDM) that provides a unified view for the existing metrics, including *Effective Diversity* (ED) [1], *Population Diversity* (PD) [42] and *Expected Cardinality* (EC) [40]. UDM can be interpreted from a geometric perspective through analyzing the meanings of the eigenvalues of a kernel matrix. Using UDM, we can also analyze the advantages and shortcomings of the existing metrics, and study why ED and PD cannot measure the diversity properly in certain cases. With our task-agnostic metric, we design the corresponding diversity-promoting objective and develop novel MARL algorithms, i.e., *UDM Fictitious Play* (UDM-FP) and *UDM Policy-Space Response Oracle* (UDM-PSRO), for solving normal-form games and open-ended games. Theoretically, we prove that our algorithms can enlarge the *gamescape* [1], i.e., the convex hull of the row vectors in the payoff matrix, by increasing the response capacity of the strategy pool, and can converge on two-player games. We validate our diversity-aware algorithms on games that show strong non-transitivity including the real-world matrix games, Blotto and non-transitive mixture model, and the empirical results show that our algorithms achieve lower *exploitability* [23] and higher *population effectivity* [30] than the baselines.

## 2 Preliminary

### 2.1 Concepts of Games

**Normal-Form Games and Open-Ended Meta-Games.** *Normal-form games* (NFGs) are denoted by $\langle N, \mathbb{S}, \boldsymbol{G} \rangle$, where $N$ is the number of players. Each player $n \in [N]$ has a finite set of pure strategies $\mathbb{S}^n$. Denote the space of joint pure-strategy profiles as $\mathbb{S} = \prod_{n \in [N]} \mathbb{S}^n$, and the space

of joint strategy profiles except the $n$-th player as $\mathbb{S}^{-n}$. $\boldsymbol{G}(S) = (\boldsymbol{G}^1(S), \cdots, \boldsymbol{G}^N(S)) \in \mathbb{R}^N$ is a payoff table mapping each joint strategy profile $S \in \mathbb{S}$ to a vector of reward values for each player. A mixed strategy (also called policy) of player $n$ is denoted by $\boldsymbol{\pi}^n \in \Delta_{\mathbb{S}^n}$, where $\Delta$ is a probability simplex. $\boldsymbol{\pi}(S) = \prod_{n \in [N]} \boldsymbol{\pi}^n(S)$ represents the probability of joint strategy profile $S$. If all players follow $\boldsymbol{\pi}$, the expected payoff of player $n$ is denoted as $\boldsymbol{G}^n(\boldsymbol{\pi}) = \boldsymbol{G}(\boldsymbol{\pi}^n, \boldsymbol{\pi}^{-n}) = \sum_{S \in \mathbb{S}} \boldsymbol{\pi}(S) \boldsymbol{G}^n(S)$.

However, it is inefficient to describe real-world games with NFGs, where the strategy space can be prohibitively large by enumerating all possible game plans [8]. For example, the number of atomic actions in StarCraft is $10^{26}$ at every time-step and the number of time-steps per episode is typically several thousands [47], in which case the number of game plans can be more than $(10^{26})^{1000}$. To describe this type of games, *meta-games* denoted by $\langle N, \mathbb{S}, \mathcal{M} \rangle$ are introduced to consider a policy as a "high-level" strategy, i.e., the *meta-strategy*. With slight abuse of notation, the space of meta-strategies is still denoted by $\mathbb{S}^n$ for each player $n$; and thus $\boldsymbol{\pi}^n \in \Delta_{\mathbb{S}^n}$ denotes the meta-policy of player $n$. Additionally, meta-games are always *open-ended* since there could be infinite meta-strategies to play a game. A meta-game payoff $\mathcal{M}$ can be described by a game engine $g : \mathbb{S} \to \mathbb{R}^n$. For example, in two-player games, the meta-game payoff can be described as $\mathcal{M} = \{g(S^1, S^2) : (S^1, S^2) \in \mathbb{S}^1 \times \mathbb{S}^2\}$, where $g(S^1, S^2) > 0$ if $S^1 \in \mathbb{S}^1$ beats $S^2 \in \mathbb{S}^2$, while $g < 0$ and $g = 0$ refer to losses and ties, respectively. A game is *symmetric* if $\mathbb{S}^1 = \mathbb{S}^2$. A game is *transitive* if there exists a monotonic rating function $h$ such that $g(S^1, S^2) = h(S^1) - h(S^2), \forall S^1 \in \mathbb{S}^1, S^2 \in \mathbb{S}^2$; and *non-transitive* if $\int_{\mathbb{S}^2} g(S^1, S^2) \mathrm{d}S^2 = 0, \forall S^1 \in \mathbb{S}^1$, which means that each strategy can be counterbalanced by another strategy.

**Solution Concepts of Games.** In a game, a (possibly mixed) strategy $\boldsymbol{\pi}^n$ is a *best response* for player $n$ against joint strategy $\boldsymbol{\pi}^{-n}$, if $\boldsymbol{\pi}^n \in \mathrm{BR}^n(\boldsymbol{\pi}^{-n})$, where $\mathrm{BR}^n(\boldsymbol{\pi}^{-n}) := \arg\max_{\tilde{\boldsymbol{\pi}} \in \Delta_{\mathbb{S}^n}}[\boldsymbol{G}^n(\tilde{\boldsymbol{\pi}}, \boldsymbol{\pi}^{-n})]$. A *Nash Equilibrium* (NE) [39] is a joint strategy $\boldsymbol{\pi}$ if $\boldsymbol{\pi}^n \in \mathrm{BR}^n(\boldsymbol{\pi}^{-n}), \forall n$. An $\epsilon$-NE is a joint strategy $\boldsymbol{\pi}$ if $\boldsymbol{\pi}^n \in \mathrm{BR}_\epsilon^n(\boldsymbol{\pi}^{-n}), \forall n$, where $\mathrm{BR}_\epsilon^n := \{\boldsymbol{\pi}^n : \boldsymbol{G}^n(\boldsymbol{\pi}^n, \boldsymbol{\pi}^{-n}) \geq \boldsymbol{G}^n(\tilde{\boldsymbol{\pi}}, \boldsymbol{\pi}^{-n}) - \epsilon\}, \tilde{\boldsymbol{\pi}} \in \mathrm{BR}^n(\boldsymbol{\pi}^{-n})$. Computing NE is PPAD-hard in N-player general-sum games [9]; and thus $\alpha$-*Rank* [41] is proposed as an alternative solution concept, which has a polynomial-time solvability in multi-player general-sum games. A more detailed description of $\alpha$-Rank can be found in Appendix A.1.

**Game Solvers.** A game solver consists of an *Oracle* function and a *(meta-)policy solver*, where the Oracle function $\mathcal{O}$ computes strategies given joint strategies, such as the best-response Oracle $\mathcal{O}^n(\boldsymbol{\pi}) = \mathrm{BR}^n(\boldsymbol{\pi}^{-n})$, and the (meta-)policy solver $\mathcal{S}$ computes the (meta)-policy based on the payoff table. At each time step, the game solver uses the policy solver to compute a policy for each player based on the current payoff; and then based on this joint policy, it uses $\mathcal{O}(\cdot)$ to find a new strategy for each player and adds it in their populations. Variations of (meta-)game solvers are summarised in Table 1.

Table 1: Various (Meta-)Game Solvers

| Method | (Meta-)Policy Solver $\mathcal{S}$ | Oracle $\mathcal{O}$ |
|---|---|---|
| Self-play [17] | $[0, \cdots, 0, 1]^N$ | $\mathrm{BR}(\cdot)$ |
| GWFP [26] | UNIFORM | $\mathrm{BR}_\epsilon(\cdot)$ |
| D.O [36] | NE | $\mathrm{BR}(\cdot)$ |
| PSRO$_N$ [23] | NE | $\mathrm{BR}_\epsilon(\cdot)$ |
| PSRO$_{rN}$ [1] | NE | $\mathrm{ED\text{-}BR}(\cdot)$ |
| $\alpha$-PSRO [38] | $\alpha$-Rank | $\mathrm{PBR}(\cdot)$ |
| EC-PSRO [40] | NE / $\alpha$-Rank | $\mathrm{EC\text{-}BR}(\cdot)$ / $\mathrm{EC\text{-}PBR}(\cdot)$ |
| Our Methods | NE / $\alpha$-Rank | Eq.(8) / Eq.(9) |

## 2.2 Evaluation Metrics

**Empirical Gamescape.** The *empirical gamescape* (EGS) [1] in a meta-game is defined as the convex hull of the payoff vectors of all meta-strategies in $\mathbb{S}^n$, written as $\mathrm{EGS}(\mathbb{S}^n) := \{\sum_i \alpha_i \cdot \boldsymbol{m}_i : \boldsymbol{\alpha} \geq 0, \boldsymbol{\alpha}^\mathsf{T} \cdot \mathbf{1} = 1, \boldsymbol{m}_i = \mathcal{M}_{[i,:]}\}$, which represents the response capacity of a pool of strategies in meta-games.

**Exploitability.** The *exploitability* [10] measures the "distance" of a joint strategy $\boldsymbol{\pi}$ to the NE, defined as $\mathrm{Exploit.}(\boldsymbol{\pi}) := \sum_{n \in [N]}[\boldsymbol{G}^n(\tilde{\boldsymbol{\pi}}, \boldsymbol{\pi}^{-n}) - \boldsymbol{G}^n(\boldsymbol{\pi})], \tilde{\boldsymbol{\pi}} \in \mathrm{BR}^n(\boldsymbol{\pi}^{-n})$.

**Population Effectivity.** The *population effectivity* (PE) [30] is a fairer evaluation metric to represent the effectiveness of a population (i.e., a strategy pool) than exploitability: $\text{PE}(\mathbb{S}^n) := \min_{\boldsymbol{\pi}^{-n}} \max_{\mathbf{1}^\mathsf{T}\boldsymbol{\alpha}=1,\alpha_k\geq 0} \sum_{k=1}^{|\mathbb{S}^n|} \alpha_k \boldsymbol{G}^n(S_k^n, \boldsymbol{\pi}^{-n})$.

### 2.3 Existing Diversity Measures

There are various diversity metrics to measure the diversity of a population.

**Effective Diversity.** *Effective Diversity* (ED) [1] is a metric proposed to measure the diversity of a pool of effective-strategies (strategies with support under the NE) in two-player zero-sum symmetric games as follows:

$$\text{ED}(\mathbb{S}^n) := \boldsymbol{\pi}^{*\mathsf{T}} \lfloor \mathcal{M} \rfloor_+ \boldsymbol{\pi}^*, \tag{1}$$

where $\lfloor x \rfloor_+ = x$ if $x > 0$ and $\lfloor x \rfloor_+ = 0$ if $x \leq 0$. $\mathcal{M}$ is the meta-payoff table of $\mathbb{S}^n$ and $\boldsymbol{\pi}^*$ is the NE of the $\mathcal{M}$.

**Expected Cardinality.** Inspired by the derterminantal point process (DPP) [31, 22], the diversity of a population can be increased through increasing the *Expected Cardinality* (EC) [40] of the random set $\mathbf{Y} \subseteq \mathbb{S}^n$ drawn from the population:

$$\text{EC}(\mathbb{S}^n) := \mathbb{E}_{\mathbf{Y}\sim\mathbb{P}_{\mathcal{L}^n}}[|\mathbf{Y}|] = \text{Tr}(\mathbf{I} - (\mathcal{L}^n + \mathbf{I})^{-1}), \tag{2}$$

where $|\mathbf{Y}|$ means the cardinality of $\mathbf{Y}$, and $\mathcal{L}^n = \mathcal{M}\mathcal{M}^\mathsf{T}$.

**Form of Euclidean Projection.** To characterize the contribution of a payoff vector to the enlargement of the EGS directly, *Form of Euclidean Projection* (FEP) [30] is defined as the form of Euclidean projection:

$$\text{FEP}(S_{\text{new}}^n) := \min_{\mathbf{1}^\mathsf{T}\boldsymbol{\beta}=1,\boldsymbol{\beta}\geq 0} \|\mathcal{M}^\mathsf{T}\boldsymbol{\beta} - \boldsymbol{a}_{\text{new}}\|_2^2, \tag{3}$$

where $S_{new}^n$ is the new meta-strategy of player $n$, and $\boldsymbol{a}_{\text{new}}^\mathsf{T} := (g^n(S_{\text{new}}^n, S_j^{-n}))_j$. Note that FEP only measures the contribution of the new meta-strategy to the current population, rather than the diversity of the population.

**Population Diversity.** *Population Diversity* (PD) [42] uses the determinant to measure the diversity by:

$$\text{PD}(\mathbb{S}^n) := \det(\mathbf{K}^n), \tag{4}$$

where $\mathbf{K}_{i,j}^n = K(\phi_i^n, \phi_j^n)$, and $\phi_i^n = \{\pi_i^n(\cdot|s)\}_s$ ($s$ means the state in games) is the behavioral embedding of a meta-strategy. Here we rename PD as *Reward-PD* (RPD) when it replaces the behavioral embedding with $\phi_i^n = \mathcal{M}_{[i,:]}$.

## 3 Methods

Based on the existing metrics, we now offer a unified view for them by introducing a novel diversity measure. Specially, this measure is based on a geometric interpretation of the diversity of strategies, and is capable of offering a comprehensive analysis of these existing diversity metrics.

### 3.1 A Unified Diversity Measure

There are various methods to represent a strategy in the existing studies. In RD, a fundamental way to represent a strategy is through the row vector of the empirical payoff matrix [40, 30], since each row in this matrix embeds the response of the corresponding strategy against different opponents; and in BD, the trajectory or the action-state distribution is often used to characterize the corresponding strategy [42, 30]. To represent a strategy more flexibly, we introduce the *strategy feature*:

**Definition 1** (Strategy Feature)**.** *Let $S_i^n \in \mathbb{S}^n$ denote the $i$-th (meta-)strategy for player $n$, then the **strategy feature** of $S_i^n$ is defined as a vector : $\phi_i^n \in \mathbb{R}^{1\times p}, p \leq M =: |\mathbb{S}^n|$, such that $\phi_i^n = \phi_j^n \iff S_i^n = S_j^n$, where $\forall S_i^n, S_j^n \in \mathbb{S}^n$.*

Hence, we can represent the $i$-th strategy by $\boldsymbol{\phi}_i^n = \boldsymbol{m}_i =: \mathcal{M}_{[i,:]}$ in RD, or $\boldsymbol{\phi}_i^n = \left\{\boldsymbol{\pi}_i^n(\cdot|s)\right\}_s$ in BD. Equipped with the strategy feature, we can then define the *diversity kernel* to measure the pairwise similarity as follows:

**Definition 2** (Diversity Kernel). *Consider a finite population $\mathbb{S}^n$ consisting of $M$ (meta-)strategies. The **diversity kernel** of player $n$ is defined as a positive semi-definite (PSD) matrix: $\mathcal{L}_K^n :=$ $[K(\boldsymbol{\phi}_i^n, \boldsymbol{\phi}_j^n)]_{M \times M}$, where $K : \mathbb{R}^d \times \mathbb{R}^d \to \mathbb{R}$ is a given kernel function, such that $|K(\boldsymbol{\phi}_i^n, \boldsymbol{\phi}_j^n)| \leq C$ and $K(\boldsymbol{\phi}_i^n, \boldsymbol{\phi}_i^n) = C$, $C > 0$.*

Given a PSD matrix $\mathcal{L} = \boldsymbol{B}\boldsymbol{B}^\mathsf{T}$, where $\boldsymbol{B} \in \mathbb{R}^{D_1 \times D_2}$ and $D_1 \geq D_2$. The geometric interpretation of $\det(\mathcal{L})$ is the squared volume of the parallelepiped spanned by the rows of $\boldsymbol{B}$ [22]. Further more, the determinant of diversity kernel represents the squared volume of a parallelepiped spanned by strategy feature corresponding to the kernel choice [42]. Thus, a population can be considered diverse if its parallelepiped can fill the strategy space as much as possible. A natural idea to measure the diversity of a population is to use the determinant of the diversity kernel, which is the intuition of PD (RPD) introduced in section 2.3. However, since the determinant becomes zero with duplicated rows, PD (RPD) cannot deal with the redundant-strategy problem which turns out to be a critical challenge for game evaluation [1] (See more discussions in Appendix A.2.). To avoid this problem, we use the additive term over eigenvalues rather than the product of eigenvalues (i.e., the determinant) to measure the diversity and propose the *Unified Diversity Measure* as follows:

**Definition 3** (Unified Diversity Measure). *Consider a function*

$$f \in \boldsymbol{F} := \left\{ f : f(x) = \sum_{k=0}^{\infty} c_k x^k, f'(x) > 0, x \in R \right\},$$

*where $R$ is the convergence domain of $f$. Denote the eigenvalues of $\mathcal{L}_K^n$ constructed by $\mathbb{S}^n = \left\{S_1^n, \cdots, S_M^n\right\}$ as $\lambda_i \geq 0$. Then the **Unified Diversity Measure** (UDM) of the population $\mathbb{S}^n$ is defined as follows:*

$$\mathrm{UDM}(\mathbb{S}^n) := \sum_{i=1}^{M} f(\lambda_i). \tag{5}$$

Since the values and the number of the non-zero eigenvalues represent the lengths and the number of the edges of the parallelepiped, respectively, increasing UDM can help enlarge the volume of the parallelepiped, which indicates a more diverse strategy population. Otherwise, considering only the number of non-zero eigenvalues (i.e., the rank of the diversity kernel) will result in the *difference* instead of the *diversity* being measured [40, 56]. For example, in RPS, the meta-strategy [0.99 Rock, 0.01 Paper] is different from [0.98 Rock, 0.02 Paper], but they are not diverse since they both favor playing Rock.

Meanwhile, we offer an equivalent representation of UDM which can be computed more easily:

**Proposition 1** (Equivalent Representation of UDM). *The UDM defined in Definition 3 has an equivalent representation:*

$$\mathrm{UDM}(\mathbb{S}^n) := \sum_{i=1}^{M} f(\lambda_i) = \mathrm{Tr}(f(\mathcal{L}_K^n)), \tag{6}$$

*where the definition of $f(\mathcal{L}_K^n)$ is similar to the matrix exponentials.*

*Proof.* See Appendix A.3.1. $\square$

Using $f(\lambda_i)$ rather than $\lambda_i$ directly is vital for UDM. In the case where $f(\lambda_i) = \lambda_i$, UDM will degenerate to $\mathrm{Tr}(\mathcal{L}^n) = \sum_i \mathcal{L}_{ii}^n$, where $\mathcal{L}_{ii}^n = K(\boldsymbol{\phi}_i^n, \boldsymbol{\phi}_i^n)$ only captures the modulus of $S_i^n$ and thus $\sum_i \mathcal{L}_{ii}^n$ ignores the similarity $\mathcal{L}_{ij}^n$ between $S_i^n$ and $S_j^n, i \neq j$, making it unable to measure the diversity properly. Instead, using bounded concave functions $f(\lambda_i)$ that involve the pairwise similarity, such as $f(\lambda_i) = \frac{1}{1+\gamma \exp(-\lambda_i)} - \frac{1}{1+\gamma}, \gamma > 0$ (the term $\frac{1}{1+\gamma}$ is applied to satisfy $f(0) = 0$), can force the player to explore more novel strategies since this function with a supremum gains more marginal benefit from adding a new non-zero eigenvalue rather than increasing an already large eigenvalue. Given the convexity of $f(x)$, UDM becomes concave due to the following proposition:

**Proposition 2** (Convexity of UDM). *Consider a concave function $f \in \boldsymbol{F}$. Then UDM is concave if all the eigenvalues of $\mathcal{L}_K^n$ exist in the convergence domain of $f$.*

*Proof.* See Appendix A.3.2. $\square$

## 3.2 Unify Existing Metrics into UDM

By mathematically transformations, we argue that some existing diversity metrics, especially the RD metrics, are the special cases of UDM, which are summarized in Table 2.

Table 2: Unify Existing Metrics into UDM

| Methods | Kernel Function $K(\cdot, \cdot)$ | Function $f$ | Strategy Feature $\boldsymbol{\phi}_i$ |
|---|---|---|---|
| ED [1] | Linear Kernel | $f(x) = x$ | $\boldsymbol{m}_i^*$ |
| PD [42] | self-selected | $f(x) = \ln x$ | $\{\boldsymbol{\pi}(\cdot\|s)\}_s$ |
| RPD | self-selected | $f(x) = \ln x$ | $\boldsymbol{m}_i$ |
| EC [40] | Linear Kernel | $f(x) = \frac{x}{1+x}$ | $\boldsymbol{m}_i$ |

**UDM vs. ED.** Formally, ED is equivalent to the $L_{1,1}$ norm of the payoff matrix in two-player symmetric zero-sum games [1]:

$$\mathrm{ED}(\mathbb{S}^n) = \boldsymbol{\pi}^{*\mathsf{T}} \lfloor \mathcal{M} \rfloor_+ \boldsymbol{\pi}^* = \frac{1}{2} \| \boldsymbol{\pi}^* \odot \mathcal{M} \odot \boldsymbol{\pi}^{*\mathsf{T}} \|_{1,1},$$

where $\boldsymbol{\pi}^*$ is the NE on $\mathcal{M}$, $\odot$ is the Hadamard product (i.e., element-wise product), and $\|A\|_{1,1} := \sum_{i,j} |a_{i,j}|$. Denote $\mathcal{M}^* := \boldsymbol{\pi}^* \odot \mathcal{M} \odot \boldsymbol{\pi}^{*\mathsf{T}}$, then:

$$\max \mathrm{ED} = \max \frac{1}{2} \| \mathcal{M}^* \|_{1,1} = \max \frac{1}{2} \sum_{i,j} |\boldsymbol{\pi}_i^* \mathcal{M}_{i,j} \boldsymbol{\pi}_j^*|$$

$$\Longleftrightarrow \max \sum_{i,j} |\boldsymbol{\pi}_i^* \mathcal{M}_{i,j} \boldsymbol{\pi}_j^*|^2 =: \max \| \mathcal{M}^* \|_F^2 = \max \sum_{i=1}^M \lambda_i^*,$$

where $\lambda_i^*$ are the eigenvalues of $\mathcal{M}^* \mathcal{M}^{*\mathsf{T}}$. Hence, ED can be covered by UDM with $K(\cdot, \cdot) = \langle \cdot, \cdot \rangle$ and $f(x) = x$. Using the strategy feature $\boldsymbol{\phi}_i = \boldsymbol{m}_i^* =: \mathcal{M}_{[i,:]}^*$ and function $f(x) = x$ in ED encourages the player to amplify its strengths and ignore its weaknesses in finding a new meta-strategy, by focusing only on the strong strategies. However, it can sometimes be problematic since weak strategies may be useful to tackle niche tasks and discover stronger strategies later during training [40]. A counter example that fails ED is the RPS-X game [38]:

$$\boldsymbol{G} = \begin{bmatrix} 0 & 1 & -1 & -2/5 \\ -1 & 0 & 1 & -2/5 \\ 1 & -1 & 0 & -2/5 \\ 2/5 & 2/5 & 2/5 & 0 \end{bmatrix}.$$

In RPS-X, if the initial population is either $\{R\}$, $\{P\}$ or $\{S\}$, then PSRO-rN [1], which is a variation of PSRO equipped with ED, will terminate in $\{R, P, S\}$ without finding $\{X\}$, since the best response of $\{R, P, S\}$ is still in $\{R, P, S\}$. However, this population will be exploited by others using strategy $X$ endlessly at test time. We show that how our method tackles this problem in Appendix A.2.

**UDM vs. PD (RPD).** We consider the case that $\mathcal{L}_K^n$ has full rank here since PD (RPD) does not work when the rank of diversity kernel is not full. Then PD (RPD) is actually UDM with $f(x) = \ln(x)$ and the linear kernel function:

$$\max \mathrm{PD} = \max \det(\mathcal{L}_K^n) = \max \prod_{i=1}^M \lambda_i \Longleftrightarrow \max \sum_{i=1}^M \ln(\lambda_i).$$

When the new eigenvalue is less than 1, PD (RPD) would decrease and thus may ignore some weak but useful meta-strategies. In fact, through the Eq. (6) and the Jacobi formula [32]: $\det(\exp(\mathbf{A})) = \exp \mathrm{Tr}(\mathbf{A})$, $\mathbf{A} \in \mathbb{R}^{M \times M}$, we can find that UDM also uses a determinant to measure the diversity. However, UDM is different from PD (RPD) since the exponential of the diversity kernel is non-decreasing with duplicated rows, and thus UDM is well-defined when dealing with the redundant-strategy problem. We give an example to illustrate how UDM tackles this issue in Appendix A.2.

**UDM vs. EC.**  EC can be rewritten as:

$$\text{EC} = \mathbb{E}_{Y \sim \mathbb{P}_{\mathcal{L}}}[|Y|] = \text{Tr}(\mathbf{I} - (\mathcal{L}^n + \mathbf{I})^{-1}) = \text{Tr}(\mathbf{I}) - \text{Tr}((\mathcal{L}^n + \mathbf{I})^{-1})$$

$$= M - \sum_{i=1}^{M} \frac{1}{1 + \lambda_1} = \sum_{i=1}^{M} \frac{\lambda_i}{1 + \lambda_i},$$

where $\lambda_i$ are the eigenvalues of $\mathcal{L}^n := \mathcal{M}\mathcal{M}^{\mathsf{T}}$. Hence, EC is a special case of QDM when $f(x) = x/(1+x)$, $K(\cdot, \cdot) = \langle \cdot, \cdot \rangle$ and strategy feature $\phi_i = \boldsymbol{m}_i$.

To summarize, the existing diversity metrics are special cases of UDM with explicit feature $\phi_i^n$, kernel function $K$, and function $f$. By exploring novel settings of these components, we can then propose a more reasonable metric to circumvent the shortcomings of existing diversity metrics.

# 4  Algorithms

Inspired by [40], we extend the classic FP, PSRO and $\alpha$-PSRO to their diverse versions by incorporating UDM into their Oracle functions, and prove their convergences in two-player games.

## 4.1  UDM Fictitious Play

At each iteration $t$, *UDM-FP* discovers a new strategy that gains a higher payoff and at the same time enriches the current population. Formally, UDM-FP only modifies the best response as follows:

$$\text{BR}_{\tau_t}^n(\boldsymbol{\pi}_t^{-n}) = \underset{\tilde{\boldsymbol{\pi}} \in \Delta_{\mathbb{S}_t^n}}{\arg\max}[\boldsymbol{G}^n(\tilde{\boldsymbol{\pi}}, \boldsymbol{\pi}_t^{-n}) + \tau_t \cdot \text{UDM}(\mathbb{S}_t^n \cup \{\tilde{\boldsymbol{\pi}}\})], \tag{7}$$

where $\tau_t$ is tunable constant; and the population is updated by adding the new strategy $\boldsymbol{\pi}^n \in \text{BR}_{\tau_t}^n(\boldsymbol{\pi}_t^{-n})$ so as $\mathbb{S}_{t+1}^n \leftarrow \mathbb{S}_t^n \cup \{\boldsymbol{\pi}^n\}$.

Intuitively, as $t \to \infty$, UDM-FP will almost surely converge to *generalised weakened fictitious play* (GWFP) [26] as long as $\tau_t \to 0$, and thus it has the same convergence guarantees with GWFP which converges to the NE on two-player zero-sum games or potential games (A more detailed description of GWFP can be found in Appendix A.1). So, we have the following proposition:

**Proposition 3** (Convergence of UDM-FP). *If UDM is concave, and UDM-FP uses the update rule:*

$$\boldsymbol{\pi}_{t+1}^n \in (1 - \alpha_{t+1})\boldsymbol{\pi}_t^n + \alpha_t(BR_{\tau_t}^n(\boldsymbol{\pi}_t^{-n}) + \boldsymbol{U}_{t+1}^n),$$

*where $\alpha_t = o(1/\log t)$ is deterministic and perturbations $\boldsymbol{U}_{t+1}^n$ are the differences between the actual and expected changes in strategies. Then UDM-FP shares the same convergence property as GWFP: the policy sequence $\boldsymbol{\pi}_t^n$ converges to the NE on two-player zero-sum games or potential games.*

*Proof.* See Appendix A.3.3 $\qquad\qquad\qquad\qquad\qquad\qquad\qquad\qquad\qquad\qquad\qquad\qquad\quad \square$

## 4.2  UDM Policy-Space Response Oracle

In open-ended (meta-)games, PSRO [23, 33, 48, 58, 35] is a more effective method than the direct search methods since there are infinite (meta-)strategies. Here we extend our diversity metric to PSRO and develop *UDM-PSRO* for open-ended (meta-)games. Suppose that player $n$ has learned a population $\mathbb{S}_t^n$ at $t$-th iteration. The goal of UDM-PSRO is to find a new (meta-)strategy $S_{\boldsymbol{\theta}}$ parameterised by $\boldsymbol{\theta} \in \mathbb{R}^d$, which maximizes the payoffs of player $n$ and the diversity of $\mathbb{S}_t^n$. Therefore, the Oracle function is:

$$\mathcal{O}^n(\boldsymbol{\pi}^{-n}) = \underset{\boldsymbol{\theta} \in \mathbb{R}^d}{\arg\max}[\sum_{S^{-n} \in \mathbb{S}^{-n}} \boldsymbol{\pi}^{-n}(S^{-n}) \cdot g(S_{\boldsymbol{\theta}}, S^{-n}) + \tau \cdot \text{UDM}(\mathbb{S}^n \cup \{S_{\boldsymbol{\theta}}\})], \tag{8}$$

where $\boldsymbol{\pi}^{-n}$ is the (meta-)policy of the player $-n$. (Meta-)policy $\boldsymbol{\pi}^n$ is computed by (meta-)policy solver, such as NE, UNIFORM, etc.

UDM-PSRO can enlarge EGS when it adds a new (meta-)strategy via Eq. (8); and thus the population learned by UDM-PSRO has a great response capacity. Unlike UDM-PSRO, PSRO-rN can also enlarge EGS, but it needs to assume that both players should use their Nash policies.

**Proposition 4** (EGS Enlargement). *Adding a new (meta-)strategy $S_{\boldsymbol{\theta}}$ via Eq. (8) enlarges EGS. Formally, we have* $\mathrm{EGS}(\mathbb{S}^n) \subseteq \mathrm{EGS}(\mathbb{S} \cup \{S_{\boldsymbol{\theta}}\})$.

*Proof.* See Appendix A.3.4. □

### 4.3 UDM $\alpha$-Policy-Space Response Oracle

UDM-PSRO suffers from the shortcomings of high complexity (i.e., PPAD-hard) in computing the NE in $N$-player general-sum games [9, 11] and the equilibrium-selection problems [19, 18]. To avoid these problems, *UDM $\alpha$-PSRO* replaces NE with $\alpha$-Rank [54] in the meta-policy solver. Suppose that there are $L$ *sink strongly-connected components* (SSCC) nodes that have only incoming edges but no outgoing edges in the response graph of the game; and denote the $\alpha$-Rank distribution of $l$-th SSCC as $\boldsymbol{\pi}^{(l)}$, which can be considered as the meta-policies for different types of opponents. Inspired by EC $\alpha$-PSRO [40], we design an Oracle function that suits $\alpha$-Rank for UDM $\alpha$-PSRO:

$$\mathcal{O}_t^n(\boldsymbol{\pi}_t^{(l)}) = \underset{\tilde{\boldsymbol{\pi}} \in \Delta_{\mathbb{S}_t^n}}{\arg\max}\, \mathrm{UDM}(\mathbb{S}_t^n \cup \{\tilde{\boldsymbol{\pi}}\}) = \underset{\tilde{\boldsymbol{\pi}} \in \Delta_{\mathbb{S}^n}}{\arg\max}\, \mathrm{Tr}\left(f(\mathcal{L}^n)\right), \tag{9}$$

where the diversity kernel $\mathcal{L}^n = [q_i K(\boldsymbol{\phi}_i, \boldsymbol{\phi}_j) q_j]_{M \times M}$ is constructed by the strategy feature $\boldsymbol{\phi}_i = \boldsymbol{m}_i / \|\boldsymbol{m}_i\|$ and the *quality term* $q_i$:

$$q_i = \exp\left\{ \mathbb{E}_{S \sim \boldsymbol{\pi}^{(l)}}\{\mathbb{1}[\mathcal{M}^n(\sigma_i^n, S^{-n}) > \mathcal{M}^n(S^n, S^{-n})]\}\right\}.$$

In fact, Eq. (9) celebrates two aspects of diversity. The first aspect is from UDM that forces players to find more diverse populations; and the second aspect is from $L$ meta-polices $\boldsymbol{\pi}^{(l)}$, which represents diverse opponents.

The following proposition shows the convergence of UDM on two-player symmetric NFGs.

**Proposition 5** (Convergence of UDM $\alpha$-PSRO). *In two-player symmetric NFGs, UDM $\alpha$-PSRO converges to the sub-cycle of the unique SSCC.*

*Proof.* See Appendix A.3.5 □

## 5 Experiments

To validate the diverse game solvers proposed in Section 4, we compare them with the baselines including self-play [17], PSRO [23], Pipeline-PSRO [34], PSRO-rN [1], FEP-PSRO [30], and EC-PSRO [40]. Their performances are investigated in several games involving both transitive and non-transitive dynamics. If an algorithm cannot find a diverse and effective population, it will easily stuck in local strategy cycles, and thus be exploited by others. Hence, exploitability and PE are used as evaluation metrics in the experiments. We focus on RD metrics and thus use the strategy feature $\boldsymbol{\phi}_i^n = \boldsymbol{m}_i$, and use the kernel function $K\langle x, y \rangle = (\langle x, y \rangle + 1)^3$ since the number of features is the same with the number of strategies. The function in UDM-PSRO is a concave function $f(x) = \frac{1}{1 + \exp(-x)} - \frac{1}{2} \in \boldsymbol{F}$ (see Appendix A.5 for more discussions on the selection of $f$ and $K$). More experimental settings and results can be found in Appendix A.4.

**Real-World Meta-Game.** The properties of some complex real-world meta-games are studied in paper [8], including AlphaStar and AlphaGO. In Figure 1, we report the performances of different algorithms over the AlphaStar game, which contains the meta-payoffs for 888 RL meta-strategies. The results show that our method achieves the smallest exploitability, largest population effectivity and the largest diversity, while baselines without diverse objectives are easily exploited since they do not find diverse meta-strategies. Our method performs better since more diverse meta-strategies can help it jump out the current strategic cycles and thus find better strategies.

**Blotto.** Blotto is a classical resource allocation game that is often used to analyze electoral competition [45]. In this game, each of two players has a budget of coins which will be distributed simultaneously over a fixed number of areas. A player wins an area where it puts the most coins; and the player that wins the most areas wins the game. We show the performances of different algorithms in this game with 3 areas and 10 coins in Figure 1; and the results show that our methods perform as well as the diversity-aware baselines, and outperforms baselines without diversity objectives in terms of exploitability and PE.

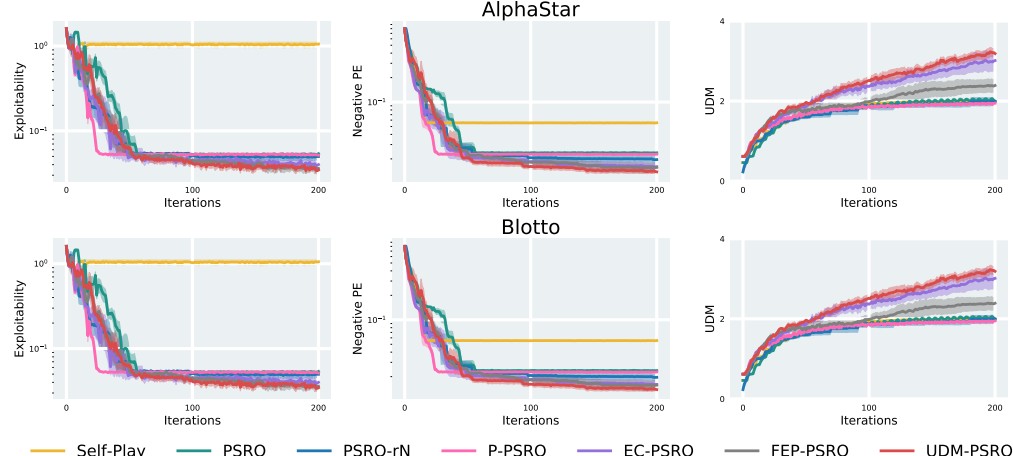

Figure 1: AlphaStar and Blotto: Exploitability & Negative PE & UDM vs. Iterations.

**Non-Transitive Mixture Model.**    Non-Transitive Mixture Model [40] is a zero-sum two-player game consisting of seven equally-distanced Gaussian humps on the 2D plane. Each meta-strategy corresponds to a point on the plane, and each point is translated into a 7-dimensional vector $S_i^n$ with each coordinate being the density in the corresponding Gaussian distribution function. Since the number of points is infinite, this game is open-ended, and it involves both a transitive component and a non-transitive component, which can be reflected by the payoff:

$$
\boldsymbol{\pi}^{1,\mathsf{T}} \begin{pmatrix} 0 & 1 & 1 & 1 & -1 & -1 & -1 \\ -1 & 0 & 1 & 1 & 1 & -1 & -1 \\ -1 & -1 & 0 & 1 & 1 & 1 & -1 \\ -1 & -1 & -1 & 0 & 1 & 1 & 1 \\ 1 & -1 & -1 & -1 & 0 & 1 & 1 \\ 1 & 1 & -1 & -1 & -1 & 0 & 1 \\ 1 & 1 & 1 & -1 & -1 & -1 & 0 \end{pmatrix} \boldsymbol{\pi}^2 + \frac{1}{2}\sum_{k=1}^{7}(\boldsymbol{\pi}_k^1 - \boldsymbol{\pi}_k^2).
$$

Therefore, a player should stay close to the center of the Gaussian and explore all the Gaussian distributions equally to avoid being exploited. We report the exploration trajectories when solving this game using different algorithms in Figure 2, which shows that our method can indeed generate diverse trajectories. We also report the exploitability and PE values for the final population generated by different algorithms in Table 3. It can be found that our method outperforms all the baselines in terms of PE, which is a better metric to evaluate diverse populations [30].

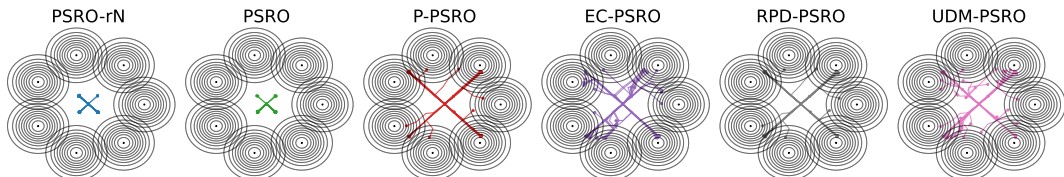

Figure 2: Non-Transitive Mixture Model: Exploration trajectories during training.

Table 3:  The OS (Opponent Strength) associated with the PE$\times 10^2$ represents the strength of the opponent during the process of using PSRO to solve it. The last row is Exploit.$\times 10^2$.

| PE(OS) | PSRO | PSRO-rN | P-PSRO | EC-PSRO | RPD-PSRO | UDM-PSRO |
|---|---|---|---|---|---|---|
| PE(10) | $-18.18 \pm 0.32$ | $-18.18 \pm 0.32$ | $9.62 \pm 0.16$ | $9.34 \pm 0.13$ | $9.68 \pm 0.19$ | $\mathbf{9.73 \pm 0.24}$ |
| PE(15) | $-27.28 \pm 0.04$ | $-27.28 \pm 0.04$ | $0.43 \pm 0.04$ | $0.01 \pm 0.04$ | $0.39 \pm 0.05$ | $\mathbf{0.44 \pm 0.06}$ |
| PE(20) | $-26.73 \pm 0.04$ | $-26.73 \pm 0.04$ | $0.10 \pm 0.12$ | $0.18 \pm 0.06$ | $0.34 \pm 0.16$ | $\mathbf{0.69 \pm 0.09}$ |
| PE(25) | $-25.47 \pm 0.07$ | $-25.47 \pm 0.07$ | $1.12 \pm 0.10$ | $1.25 \pm 0.17$ | $1.39 \pm 0.14$ | $\mathbf{1.81 \pm 0.18}$ |
| Exploit. | $33.71 \pm 0.37$ | $35.11 \pm 0.23$ | $2.34 \pm 0.43$ | $2.05 \pm 0.38$ | $\mathbf{1.95 \pm 0.54}$ | $2.07 \pm 0.37$ |

# 6  Conclusion

In this paper, we offer a consistent definition for the diversity of strategies in MARL, and propose a novel diversity measure that provides a unified view for existing diversity metric. With this measure, we develop corresponding diversity-promoting algorithms based on FP and PSRO. In theory, we prove that our method converges to two-player NE and can enlarge gamescape by increasing the response capacity of the population. Empirically, we validate our method on games that show strong non-transitive including the matrix games, Blotto and non-transitive mixture model. The results show that our method outperforms the baselines in terms of the exploitability and population effectivity. In this paper, we mainly focus on the two-player games due to the expensive computational cost of UDM in n-player cases (see Appendix 5 for more explanations). Investigating how to reduce the computational cost when extending UDM to n-player, general-sum, or non-symmetric games can be an important future work.

## Acknowledgments and Disclosure of Funding

The authors would like to thank anonymous reviewers for helpful feedback and discussions. This work was supported by the SYSU-ByteDance Research Project, and the National Natural Science Foundation of China under Grant 62076259.

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
