# Supplementary Material for
# A Unified Diversity Measure for Multiagent Reinforcement Learning

## Contents

# A  Appendix

## A.1  Notations and Additional Concepts

### A.1.1  Notations

Table 1: Notations.

| Name | Notation | Meaning |
|------|----------|---------|
| strategy | $S^n$ | the strategy selected by player $n$ |
| policy | $\boldsymbol{\pi}^n$ | the tool of player $n$ used to select strategies |
| game engine | $g : \mathbb{S} \to \mathbb{R}^n$ | compute the utility of a joint strategy |
| best response | $BR^n$ | the strategies with the highest utility |
| Oracle | $\mathcal{O}^n$ | give a response for player $n$ |
| policy solver | $\mathcal{S}^n$ | compute a policy for player $n$ |
| rectifier | $\lfloor x \rfloor$ | $\lfloor x \rfloor_+ := \max(0, x)$ |
| Hadamard product | $\odot$ | $[a_{ij}]_{m \times n} \odot [b_{ij}]_{m \times n} := [a_{ij} b_{ij}]_{m \times n}$ |
| 1,1-norm | $\|\cdot\|_{1,1}$ | $\|A\|_{1,1} := \sum_{ij} |a_{ij}|$ |
| Frobenius norm | $\|\cdot\|_F$ | $\|A\|_F := \sqrt{\sum_{ij} a_{ij}^2} = \sqrt{\mathrm{Tr}(A^\mathsf{T} A)}$ |
| cardinality of a finite set | $|\cdot|$ | the number of elements in a finite set |
| **Evaluation Metrics** | | |
| empirical gamescape | $\mathrm{EGS}(\mathbb{S}^n)$ | a convex hull of the payoff vectors of strategies in $\mathbb{S}^n$ |
| exploitability | $\mathrm{Exploit.}(\boldsymbol{\pi})$ | measure the "distance" of a joint policy to the NE |
| population effectivity | $\mathrm{PE}(\mathbb{S}^n)$ | represent the effectiveness of a population |
| **Existing Diversity Measures** | | |
| effective diversity | $\mathrm{ED}(\mathbb{S}^n)$ | a metric to measure the diversity of $\mathbb{S}^n$ |
| expected cardinality | $\mathrm{EC}(\mathbb{S}^n)$ | a metric to measure the diversity of $\mathbb{S}^n$ |
| population diversity | $\mathrm{PD}(\mathbb{S}^n)$ | a metric to measure the diversity of $\mathbb{S}^n$ |
| form of Euclidean projection | $\mathrm{FEP}(S^n_{\mathrm{new}})$ | compute the contribution of $S^n_{\mathrm{new}}$ to the current population |
| **About Our Method** | | |
| strategy feature | $\phi^n$ | characterize the corresponding strategy of player $n$ |
| diversity kernel | $\mathcal{L}^n_K$ | a PSD matrix computed by the given kernel $K$ |
| unified diversity measure | $\mathrm{UDM}(\mathbb{S}^n)$ | a metric to measure the diversity of $\mathbb{S}^n$ |

### A.1.2  $\alpha$-Rank

$\alpha$-Rank [5] is a solution concept built on the *response graph* of a game. Each strategy profile $S \in \mathbb{S}$ of the game is a node of this response graph, and a directed edge points from any profile $S \in \mathbb{S}$ to $\sigma \in \mathbb{S}$ in the graph if (1) $S$ and $\sigma$ differ in only one single player's strategy, and (2) $\boldsymbol{G}^n(\sigma) > \boldsymbol{G}^n(S)$. Thus, $\alpha$-Rank constructs a random walk along this directed graph; and by injecting a small probability of backwards-transitions from $\sigma$ to $S$ in this process, this random walk would be equivalent to irreducible Markov chain, which ensures the existence of a unique stationary distribution $\boldsymbol{\pi} \in \Delta_{\mathbb{S}^n}$ called the $\alpha$-Rank distribution. The masses of $\boldsymbol{\pi}$ are supported by the *sink strongly-connected components* (SSCC) nodes, which are the nodes with only incoming edges but no outgoing edges. More details of $\alpha$-Rank distribution can be found in paper [5].

### A.1.3  GWFP

*Generalised weakened fictitious play* (GWFP) [2] generalises *fictitious play* (FP) by allowing for $\epsilon$-best responses and perturbed averages strategy updates. Its definition is as follows:

**Definition 1** (GWFP). *GWFP is a process of $\{\boldsymbol{\pi}_t\}_{t \geq 0}$ with $\boldsymbol{\pi}_t \in \prod_{n \in [N]} \Delta_{\mathbb{S}^n}$, such that*

$$\boldsymbol{\pi}^n_{t+1} \in (1 - \alpha_{t+1})\boldsymbol{\pi}^n_t + \alpha_{t+1}(BR^n_{\epsilon_t}(\boldsymbol{\pi}^{-n}_t) + \boldsymbol{U}^n_{t+1}), \tag{1}$$

*where $\epsilon_t \to 0$ and $\alpha_t \to 0$ as $t \to \infty$, $\sum_{t \geq 1} \alpha_t = \infty$, and $\{U_t\}_{t \geq 1}$ is a sequence of perturbations such that, for any $T > 0$,*

$$\lim_{t \to \infty} \sup_k \left\{ \|\sum_{i=t}^{k-1} \alpha_{i+1} U_{i+1}\| : \sum_{i=t}^{k-1} \alpha_i < T \right\} = 0. \tag{2}$$

*GWFP recovers FP if $\alpha_t = 1/t$, $\epsilon_t = 0$ and $U_t = 0, \forall t$.*

In theory, the policy sequence $\{\pi_t\}_{t \geq 0}$ learned by GWFP converges to the NE on two-player zero-sum games or potential games.

## A.2 Some Examples

### A.2.1 RPS-X Game

The RPS-X game is a counter example where ED would ignore the weak but useful strategy $X$. To gain a conceptual understanding of why our method is able to find the final strategy $X$, consider the scenario where we are in the state of the population $\{R, P, S\}$. In this case, PSRO-rN would fail to find the final strategy $X$ since the best response strategy is still within $\{R, P, S\}$ [3]. We specify UDM with $f(x) = \frac{1}{1+\exp(-x)} - \frac{1}{2}$, linear kernel function and strategy feature $\phi_i^n = G_{[i,:]}$, then our method can add the strategy $X$ to the population with the largest UDM value as the best response. Consider the following four cases:

(a) Strategy $R$ is added to the population and then we have a strategy set $\{R, P, S, R\}$, resulting in the following payoff matrix:

$$
G = \begin{bmatrix} 0 & -1 & 1 & 0 \\ 1 & 0 & -1 & 1 \\ -1 & 1 & 0 & -1 \\ 0 & -1 & 1 & 0 \end{bmatrix},
$$

and the following diversity kernel:

$$
\mathcal{L} = GG^{\mathsf{T}} = \begin{bmatrix} 2 & -1 & -1 & 2 \\ -1 & 3 & -2 & -1 \\ -1 & -2 & 3 & -1 \\ 2 & -1 & -1 & 2 \end{bmatrix}.
$$

Then the UDM value is: $\text{UDM}(\{R, P, S, R\}) \approx 0.987$.

(b) Strategy $P$ is added to the population and then we have a strategy set $\{R, P, S, P\}$, and the result is the same as that of adding $R$ to the population and the UDM value is: $\text{UDM}(\{R, P, S, P\}) \approx 0.987$.

(c) Strategy $S$ is added to the population and then we have a strategy set $\{R, P, S, S\}$, and the result is the same as that of adding $R$ to the population and the UDM value is: $\text{UDM}(\{R, P, S, S\}) \approx 0.987$.

(d) Strategy $X$ is added to the population and then we have a strategy set $\{R, P, S, X\}$, resulting in the following payoff matrix:

$$
G = \begin{bmatrix} 0 & -1 & 1 & -2/5 \\ 1 & 0 & -1 & -2/5 \\ -1 & 1 & 0 & -2/5 \\ 2/5 & 2/5 & 2/5 & 0 \end{bmatrix},
$$

and the following diversity kernel:

$$
\mathcal{L} = GG^{\mathsf{T}} = \begin{bmatrix} 2.16 & -0.84 & -0.84 & 0 \\ -0.84 & 2.16 & -0.84 & 0 \\ -0.84 & -0.84 & 2.16 & 0 \\ 0 & 0 & 0 & 0.48 \end{bmatrix}.
$$

Then the UDM value is: $\text{UDM}(\{R, P, S, X\}) \approx 1.141$. Therefore, strategy $X$ would be added to the population due to the largest UDM value.

### A.2.2 Redundant-Strategy Problem

Here we use the Rock-Paper-Scissors (RPS) game to illustrate how our method tackles redundant-strategy problem:

$$
G_1 = \begin{bmatrix} 0 & -1 & 1 \\ 1 & 0 & -1 \\ -1 & 1 & 0 \end{bmatrix} \quad \text{and} \quad G_2 = \begin{bmatrix} 0 & -1 & 1 & 1 \\ 1 & 0 & -1 & -1 \\ -1 & 1 & 0 & 0 \\ -1 & 1 & 0 & 0 \end{bmatrix}.
$$

The first payoff table encodes rock-paper-scissors interactions, and the second is the same payoff but with two copies of scissors, and thus the diversity kernels with strategy feature $\phi_i^n = \frac{G_{[i,:]}}{\|G_{[i,:]}\|}$ can be computed as follows:

$$\mathcal{L}_1 = \begin{bmatrix} 1 & -1/2 & -1/2 \\ -1/2 & 1 & -1/2 \\ -1/2 & -1/2 & 1 \end{bmatrix} \quad \text{and} \quad \mathcal{L}_2 = \begin{bmatrix} 1 & -2/3 & -\sqrt{6}/6 & -\sqrt{6}/6 \\ -2/3 & 1 & -\sqrt{6}/6 & -\sqrt{6}/6 \\ -\sqrt{6}/6 & -\sqrt{6}/6 & 1 & 1 \\ -\sqrt{6}/6 & -\sqrt{6}/6 & 1 & 1 \end{bmatrix}.$$

Their eigenvalues are $\lambda(\mathcal{L}_1) = \{\frac{3}{2}, \frac{3}{2}, 0\}$ and $\lambda(\mathcal{L}_2) = \{\frac{7}{3}, \frac{5}{3}, 0, 0\}$, respectively, and thus RPD $= 0$ in both cases while the population reaches the largest diversity. In contrast, specifying UDM with linear kernel function and $f(x) = \frac{x}{1+x}$, UDM values can be computed as follows:

$$\text{UDM}(\{R, P, S\}) = f(\frac{3}{2}) + f(\frac{3}{2}) + f(0) = \frac{6}{5} = \frac{48}{40},$$

$$\text{UDM}(\{R, P, S, S\}) = f(\frac{7}{3}) + f(\frac{5}{3}) + f(0) + f(0) = \frac{53}{40}.$$

It can be found that UDM does not slump with a redundant strategy.

Besides, from this example, we can obtain:

$$\text{EC}(\{R, P, S\}) = \text{Tr}(\mathbf{I} - (\mathbf{I} + \mathcal{L}_1)^{-1}) = \frac{6}{5} > 1 = \text{rank}(G_1)/2. \tag{3}$$

Nieves et al. [4] argue that if $G$ is normalised (i.e., $\|G_{[i,:]}\| = 1, \forall i$), then $\text{EC}(\mathbb{S}^n) \leq \text{rank}(G)/2$, which is problematic since it conflicts with Eq. (3).

## A.3 A Full Proof of Propositions

### A.3.1 Proof of Proposition 1

**Proposition 1** (Equivalent Representation of UDM). *The UDM defined in Definition 3 has an equivalent representation:*

$$\text{UDM}(\mathbb{S}^n) := \sum_{i=1}^{M} f(\lambda_i) = \text{Tr}(f(\mathcal{L}_K^n)). \tag{4}$$

*Proof.* Since $\mathcal{L}_K^n$ is PSD, there exists an orthogonal matrix $\boldsymbol{P}$ such that:

$$\boldsymbol{P}^\mathsf{T} \mathcal{L}_K^n \boldsymbol{P} = \text{diag}\{\lambda_i\} =: \boldsymbol{\Lambda},$$

where $\lambda_i$ is the eigenvalue of $\mathcal{L}_K^n$. If $\lambda_i \in R, \forall i$, we have:

$$\begin{aligned}
f(\mathcal{L}_K^n) &= \sum_{k=0}^{\infty} c_k (\mathcal{L}_K^n)^k \\
&= \sum_{k=0}^{\infty} c_k (\boldsymbol{P} \cdot \boldsymbol{\Lambda} \cdot \boldsymbol{P}^\mathsf{T})^k \\
&= \boldsymbol{P} \cdot (\sum_{k=0}^{\infty} c_k \boldsymbol{\Lambda}^k) \cdot \boldsymbol{P}^\mathsf{T} \\
&= \boldsymbol{P} \cdot \text{diag}\{f(\lambda_i)\} \cdot \boldsymbol{P}^\mathsf{T}.
\end{aligned}$$

Hence, $f(\lambda_i), i = 1, \cdots, M$ are the eigenvalues of $f(\mathcal{L}_K^n)$, and thus $\sum_{i=1}^{M} f(\lambda_i) = \text{Tr}(f(\mathcal{L}_K^n))$, proving our proposition.

$\square$

### A.3.2 Proof of Proposition 2

Before the proof of proposition 2, we provide a lemma and its corollary.

**Lemma 1.** *Let $\boldsymbol{A}, \boldsymbol{B} = [b_{i,j}]_{n \times n} \in \mathbb{R}^{n \times n}$, where $\boldsymbol{A} = diag\{\mu_1, \cdots, \mu_n\}$. Denote the eigenvalues of $\boldsymbol{A} + \varepsilon \boldsymbol{B}$ by $\lambda_i(\varepsilon)$, where $\epsilon$ is a real number with sufficiently small $|\varepsilon|$; and thus $\lambda_i(0) = \mu_i, i = 1, \cdots, n$. Suppose that $\mu$ is the eigenvalue of $\boldsymbol{A}$ with multiplicity $m$, i.e., there exists a subsequence $J = \{l_1, \cdots, l_m\} \subseteq \{1, \cdots, n\}$ such that:*

$$\mu_{l_1} = \mu_{l_2} = \cdots = \mu_{l_m} = \mu, \quad \mu_j \neq \mu, \forall j \notin J.$$

*Hence,*

*(a)* $\sum_{i \in J} \lambda_j'(0) = \sum_{j \in J} b_{jj}$;

*(b)* $\sum_{i \in J} \lambda_j''(0) = 2 \sum_{j \notin J} \dfrac{\sum_{j \in J} b_{ij} b_{ji}}{\mu - \mu_j}$.

*Proof.* Without loss of generality, we assume that $J = \{1, \cdots, m\}$. Let

$$\tilde{\boldsymbol{A}} = \boldsymbol{A} - \mu \mathbf{I}_n = \text{diag}\{0, \cdots, 0, \tilde{\mu}_{m+1}, \cdots, \tilde{\mu}_n\}, \quad \tilde{\mu}_k = \mu_k - \mu.$$

It is easy to compute that the eigenvalues of $\tilde{\boldsymbol{A}} + \varepsilon \boldsymbol{B}$ are $\tilde{\lambda}_i(\varepsilon) = \lambda_i(\varepsilon) - \mu, \forall i$. Denote the column vectors of $\lambda \mathbf{I}_n - \tilde{\boldsymbol{A}} - \varepsilon \boldsymbol{B}$ by $\eta_i(\lambda, \varepsilon), i = 1, \cdots, n$, and $F(\lambda, \varepsilon) := \det(\lambda \mathbf{I}_n - \tilde{\boldsymbol{A}} - \varepsilon \boldsymbol{B})$. Note that $\frac{\partial^2}{\partial \varepsilon^2} \eta_i = 0$, then:

$$\frac{\partial}{\partial \varepsilon} F = \sum_{i=1}^{n} \det(\eta_1, \cdots, \eta_{i-1}, \frac{\partial}{\partial \varepsilon} \eta_i, \eta_{i+1}, \cdots, \eta_n),$$

$$\frac{\partial^2}{\partial \varepsilon^2} F = 2 \sum_{1 \leq i < j \leq n} \det(\eta_1, \cdots, \eta_{i-1}, \frac{\partial}{\partial \varepsilon} \eta_i, \eta_{i+1}, \cdots, \eta_{j-1}, \frac{\partial}{\partial \varepsilon} \eta_j, \eta_{j+1}, \cdots, \eta_n).$$

Let $\varepsilon = 0$, we have:

$$\frac{\partial}{\partial \varepsilon} F(\lambda, 0) = -\lambda^{m-1} \prod_{j=m+1}^{n} (\lambda - \tilde{\mu}_j) \cdot \sum_{i=1}^{m} b_{ii} - \lambda^m \sum_{i=m+1}^{n} b_{ii} \prod_{j \geq m+1, j \neq i} (\lambda - \tilde{\mu}_j), \qquad (5)$$

$$\begin{aligned}
\frac{\partial^2}{\partial \varepsilon^2} F(\lambda, 0) =& 2\delta(m-1) \cdot \lambda^{m-2} \Big( \sum_{1 \leq i < j \leq m} \det(B_{ij}) \Big) \prod_{k=m+1}^{n} (\lambda - \tilde{\mu}_k) \\
&+ 2\lambda^{m-1} \sum_{1 \leq i \leq m < j \leq n} [\det(B_{ij}) \prod_{k \geq m+1, k \neq j} (\lambda - \tilde{\mu}_k)] \\
&+ 2\lambda^m \sum_{m < i < j \leq n} [\det(B_{ij}) \prod_{k \geq m+1, k \neq i, j} (\lambda - \tilde{\mu}_k)],
\end{aligned} \qquad (6)$$

where

$$\delta(x) = \begin{cases} 1 & , x \neq 0 \\ 0 & , x = 0 \end{cases}, \quad B_{ij} = \begin{pmatrix} b_{ii} & b_{ij} \\ b_{ji} & b_{jj} \end{pmatrix}$$

On the other hand, note that:

$$F(\lambda, \varepsilon) \equiv \prod_{i=1}^{m} (\lambda - \tilde{\lambda}_i(\varepsilon)) \prod_{j=m+1}^{n} (\lambda - \tilde{\lambda}_j(\varepsilon)).$$

Differentiate the both sides of the above equation with respect to $\varepsilon$ and let $\varepsilon = 0$, then we have:

$$\begin{aligned}
\frac{\partial}{\partial \varepsilon} F(\lambda, 0) =& -\lambda^{m-1} \Big( \sum_{i=1}^{m} \lambda'_i(0) \Big) \prod_{j=m+1}^{n} (\lambda - \tilde{\mu}_j) \\
&- \lambda^m \sum_{i=m+1}^{n} \lambda'_i(0) \prod_{j \geq m+1, j \neq i} (\lambda - \tilde{\mu}_j),
\end{aligned} \qquad (7)$$

$$\begin{aligned}
\frac{\partial^2}{\partial \varepsilon^2} F(\lambda, 0) =& \lambda^{m-1} \left[ \Big( \sum_{i=1}^{m} \lambda''_i(0) \Big) \prod_{j=m+1}^{n} (\lambda - \tilde{\mu}_j) - 2 \sum_{i=1}^{m} \lambda'_i(0) \sum_{i=m+1}^{n} \lambda'_i(0) \prod_{j \geq m+1, j \neq i} (\lambda - \tilde{\mu}_j) \right] \\
&+ 2\delta(m-1) \cdot \lambda^{m-2} \sum_{1 \leq i < j \leq m} \lambda'_i(0)\lambda'_j(0) \prod_{k=m+1}^{n} (\lambda - \tilde{\mu}_k) + \lambda^m R(\lambda),
\end{aligned} \qquad (8)$$

where $R(\lambda)$ is a polynomial with respect to $\lambda$.

Since $\left. \dfrac{(5) - (7)}{\lambda^{m-1}} \right|_{\lambda=0} = 0$:

$$\sum_{i=1}^{m} \lambda'_i(0) = \sum_{i=1}^{m} b_{ii}. \qquad (9)$$

Now we prove the first conclusion in Lemma 1.

If $m > 1$, since $\left. \dfrac{(6) - (8)}{\lambda^{m-2}} \right|_{\lambda=0} = 0$:

$$\sum_{1 \leq i < j \leq m} \det(B_{ij}) = \sum_{1 \leq i < j \leq m} \lambda'_i(0)\lambda'_j(0). \qquad (10)$$

If $m = 1$, since $\left. \dfrac{(6) - (8)}{\lambda^{m-1}} \right|_{\lambda=0} = 0$:

$$\prod_{k=m+1}^{n} (-\tilde{\mu}_k) \cdot \sum_{i=1}^{m} \lambda''_i(0) = 2 \sum_{i=1}^{m} \sum_{j=m+1}^{n} (\lambda'_i(0)\lambda'_j(0) - \det(B_{ij})) \prod_{k \geq m+1, k \neq j} (-\tilde{\mu}_k). \qquad (11)$$

For $\forall j \geq m + 1$, if $\tilde{\mu}_j$ is the eigenvalue of $\tilde{A}$ with multiplicity $r$, we denote these eigenvalues by:

$$\tilde{\mu}_{m+1} = \cdots = \tilde{\mu}_{m+r}.$$

Besides, since the first conclusion in Lemma 1 :

$$\sum_{j=m+1}^{m+r} \lambda'_j(0) = \sum_{j=m+1}^{m+r} b_{jj},$$

then we have:

$$\sum_{i=1}^{m} \sum_{j=m+1}^{m+r} (\lambda'_i(0)\lambda'_j(0) - b_{ii}b_{jj} + b_{ij}b_{[ji]})(-\tilde{\mu}_{m+1})^{r-1} \prod_{k \geq m+r} (-\tilde{\mu}_k)$$

$$=(-\tilde{\mu}_{m+1})^{r-1} \prod_{k \geq m+r} (-\tilde{\mu}_k) \left[ \sum_{i=1}^{m}(\lambda'_i(0) \sum_{j=m+1}^{m+r} \lambda'_j(0) - b_{ii} \sum_{j=m+1}^{m+r} b_{jj}) + \sum_{1 \leq i \leq m < j \leq m+r} b_{ij}b_{ji} \right]$$

$$=(-\tilde{\mu}_{m+1})^{r-1} \prod_{k \geq m+r} (-\tilde{\mu}_k) \left[ \sum_{i=1}^{m}(\lambda'_i(0) - b_{ii}) \sum_{j=m+1}^{m+r} b_{jj} + \sum_{1 \leq i \leq m < j \leq m+r} b_{ij}b_{ji} \right]$$

$$=(-\tilde{\mu}_{m+1})^{r-1} \prod_{k \geq m+r} (-\tilde{\mu}_k) \sum_{i=1}^{m} \sum_{j=m+1}^{m+r} b_{ij}b_{ji}$$

$$=\sum_{i=1}^{m} \sum_{j=m+1}^{m+r} b_{ij}b_{ji} \prod_{k \geq m+1, k \neq j} (-\tilde{\mu}_k). \tag{12}$$

Therefore, (11) can be rewritten as:

$$\prod_{k=m+1}^{n} (-\tilde{\mu}_k) \cdot \sum_{i=1}^{m} \lambda''_i(0) = 2\sum_{i=1}^{m} \sum_{j=m+1}^{n} b_{ij}b_{ji} \prod_{k \geq m+1, k \neq j} (-\tilde{\mu}_k),$$

then we have:

$$\sum_{i=1}^{m} \lambda''_i(0) = 2 \sum_{j=m+1}^{n} \frac{\sum_{i=1}^{m} b_{ij}b_{ji}}{-\tilde{\mu}_j} = 2 \sum_{j=m+1}^{n} \frac{\sum_{i=1}^{m} b_{ij}b_{ji}}{\mu - \mu_j}.$$

Therefore, we prove the second conclusion in Lemma 1; and then we complete the proof.

$\square$

**Corollary 1.** *Let* $A = diag\{\mu_1 I_{r_1}, \cdots, \mu_m I_{r_m}\}$, *where* $\mu_1 < \mu_2 < \cdots < \mu_m$, $\sum_{i=1}^{m} r_i = n$, *and* $B = [b_{ij}]_{n \times n} \in \mathbb{R}^{n \times n}$. *Denote the eigenvalues of* $A + \varepsilon B$ *as* $\lambda_i(\varepsilon), i = 1, \cdots, m, j = 1, \cdots, r_i$, *and thus* $\lambda_{ij}(0) = \mu_i$.

*Then:*

(a) $\sum_{i=1}^{k} \sum_{j=i}^{r_i} \lambda''_{ij}(0) \leq 0, k = 1, \cdots, m - 1$;

(b) $\sum_{i=1}^{m} \sum_{j=1}^{r_i} \lambda''_{ij}(0) = 0$.

*The equal sign in the case* $k < m$ *holds if and only if* $b_{ij} = 0, \forall i = 1, \cdots, \sum_{s=1}^{k} r_s, j = \sum_{s=1}^{k} r_s + 1, \cdots, m$.

*Proof.* Let $R_0 = 0, R_i = R_{i-1} + r_i, i = 1, \cdots, m$, and

$$\beta_{st} = 2 \sum_{j=R_{s-1}+1}^{R_s} \sum_{i=R_{t-1}+1}^{R_t} b_{ij}b_{ji} = \beta_{st} = 2 \sum_{j=R_{s-1}+1}^{R_s} \sum_{i=R_{t-1}+1}^{R_t} b_{ij}^2 = \beta_{ts} \geq 0.$$

According to the second conclusion in Lemma 1, we have:

$$\sum_{j=1}^{r_i} \lambda_{ij}''(0) = \sum_{j \geq 1, j \neq i}^{m} \frac{\beta_{ij}}{\mu_i - \mu_j}.$$

Therefore,

$$\sum_{i=1}^{k} \sum_{j=1}^{r_i} \lambda_{ij}''(0) = \sum_{i=1}^{k} \sum_{j=1, j \neq i}^{m} \frac{\beta_{ij}}{\mu_i - \mu_j} = \sum_{i=1}^{k} \sum_{j=1, j \neq i}^{k} \frac{\beta_{ij}}{\mu_i - \mu_j} + \sum_{i=1}^{k} \sum_{j=k+1}^{m} \frac{\beta_{ij}}{\mu_i - \mu_j} \quad (13)$$

$$= \sum_{i=1}^{k} \sum_{j=k+1}^{m} \frac{\beta_{ij}}{\mu_i - \mu_j} \leq 0, \quad \forall k < m. \quad (14)$$

The last "=" holds if and only if:

$$\beta_{ij} = 0,$$

which is equivalent to $b_{ij} = 0, \forall i = 1, \cdots, R_k, j = R_{k+1}, \cdots, n$.

Similarly, we can prove the case where $k = m$, and thus we can complete the proof.

$\square$

**Proposition 2** (Convexity of UDM). *Consider a concave function $f \in \boldsymbol{F}$. Then UDM is concave if all the eigenvalues of $\mathcal{L}_K^n$ exist in the convergence domain of $f$.*

*Proof.* We study the sign of the second derivative of the UDM in a neighborhood of the PSD matrix $\mathcal{L}_K^n$, and we rewrite $\mathcal{L}_K^n$ as $\mathcal{L}$ for convenience. We apply a perturbation to $\mathcal{L}$ such that $\mathcal{L} + \varepsilon \boldsymbol{B}$ with a symmetric matrix $\boldsymbol{B}$ and $\varepsilon \in \mathbb{R}$. It suffices to prove that:

$$\frac{\mathrm{d}^2}{\mathrm{d}\varepsilon^2} \mathrm{Tr} f(\mathcal{L} + \varepsilon \boldsymbol{B}) \bigg|_{\varepsilon=0} < 0.$$

First, there exists an orthogonal matrix $\boldsymbol{C}$ such that:

$$\boldsymbol{\Lambda} := \boldsymbol{C}^\mathsf{T} \mathcal{L} \boldsymbol{C} = \mathrm{diag}\{\mu_1 \boldsymbol{I}_{r_1}, \cdots, \mu_m \boldsymbol{I}_{r_m}\},$$

where $\mu_1 < \cdots < \mu_m, \sum_{i=1}^{m} r_i = n$.

Besides, there exists an orthogonal matrix $\boldsymbol{P}(\varepsilon)$ such that:

$$\boldsymbol{P}^\mathsf{T}(\varepsilon)(\boldsymbol{\Lambda} + \varepsilon \boldsymbol{C}^\mathsf{T} \boldsymbol{B} \boldsymbol{C})\boldsymbol{P}(\varepsilon) = \mathrm{diag}\{\lambda_{11}(\varepsilon), \cdots, \lambda_{1r_1}(\varepsilon), \cdots, \lambda_{m1}(\varepsilon), \cdots, \lambda_{mr_m}(\varepsilon)\},$$

where $\lambda_{ij}(0) = \mu_i, j = 1, \cdots, r_i, i = 1, \cdots, m$.

Hence,

$$\frac{\mathrm{d}^2}{\mathrm{d}\varepsilon^2} \mathrm{Tr} f(\mathcal{L} + \varepsilon \boldsymbol{B}) \bigg|_{\varepsilon=0}$$

$$= \frac{\mathrm{d}^2}{\mathrm{d}\varepsilon^2} \mathrm{Tr} \left[ \boldsymbol{P}^\mathsf{T}(\varepsilon) \boldsymbol{C}^\mathsf{T} f(\mathcal{L} + \varepsilon \boldsymbol{B}) \boldsymbol{C} \boldsymbol{P}(\varepsilon) \right] \bigg|_{\varepsilon=0}$$

$$= \frac{\mathrm{d}^2}{\mathrm{d}\varepsilon^2} \mathrm{Tr} f \left( \boldsymbol{P}^\mathsf{T}(\varepsilon) \boldsymbol{C}^\mathsf{T} (\mathcal{L} + \varepsilon \boldsymbol{B}) \boldsymbol{C} \boldsymbol{P}(\varepsilon) \right) \bigg|_{\varepsilon=0}$$

$$= \frac{\mathrm{d}^2}{\mathrm{d}\varepsilon^2} \sum_{i=1}^{m} \sum_{j=1}^{r_i} f(\lambda_{ij}(\varepsilon)) \bigg|_{\varepsilon=0}$$

$$= \sum_{i=1}^{m} \left( \sum_{j=1}^{r_i} \lambda_{ij}''(0) \right) f'(\mu_i) + \sum_{i=1}^{m} \sum_{j=1}^{r_i} \left( \lambda_{ij}'(0) \right)^2 f''(\mu_i)$$

$$= \sum_{k=1}^{m-1} [f'(\mu_k) - f'(\mu_{k+1})] \sum_{i=1}^{k} \sum_{j=1}^{r_i} [\lambda_{ij}''(0)] + \sum_{i=1}^{m} \sum_{j=1}^{r_i} [\lambda_{ij}'(0)]^2 f''(\mu_i) \quad (15)$$

From Corollary 1 and $f'(\mu_k) - f'(\mu_{k+1}) > 0$, it can be easily derived that (15)$\leq 0$.

If (15)$= 0$, then $\boldsymbol{C}^\mathsf{T} \boldsymbol{B} \boldsymbol{C}$ is a diagonal block matrix according to Corollary 1. Denote its diagonal elements as $\tilde{b}_{ij}, i = 1, \cdots, m, j = 1, \cdots, r_i$ where not all of these values are zero. However, from (15)$= 0$ we have:

$$0 = \lambda'_{ij}(0) = \tilde{b}_{ij}, \forall i, j.$$

However, it contradicts $\tilde{b}_{ij}, i = 1, \cdots, m, j = 1, \cdots, r_i$ where not all of these values are zero.

Therefore, we have (15)$< 0$, yielding the desired result.

$\square$

### A.3.3 Proof of Proposition 3

**Proposition 3** (Convergence of UDM-FP). *If UDM is concave, and UDM-FP uses the update rule:*

$$\boldsymbol{\pi}^n_{t+1} \in (1 - \alpha_{t+1})\boldsymbol{\pi}^n_t + \alpha_t(BR^n_{\tau_t}(\boldsymbol{\pi}^{-n}_t) + \boldsymbol{U}^n_{t+1}),$$

*where $\alpha_t = o(1/\log t)$ is deterministic and perturbations $\boldsymbol{U}^n_{t+1}$ are the differences between the actual and expected changes in strategies. Then UDM-FP shares the same convergence property as GWFP: the policy sequence $\boldsymbol{\pi}^n_t$ converges to the NE on two-player zero-sum games or potential games.*

*Proof.* From the assumption, UDM is a concave function, and $\tau_t \to 0$ as $t \to \infty$; and perturbations are bounded martingale differences since they are the differences between the actual and expected change in strategies. So if $\{\alpha_t\}_{t \geq 1}$ is deterministic and $\alpha_t = o(1/\log t)$, then for $\forall T > 0$, the condition on $\boldsymbol{U}^n_{t+1}$, i.e.:

$$\mathbb{P}\left\{ \lim_{t \to \infty} \sup_k \left\{ \|\sum_{i=t}^{k-1} \alpha_{i+1}\boldsymbol{U}_{i+1}\| : \sum_{i=t}^{k-1} \alpha_i < T \right\} = 0 \right\} = 1$$

holds with probability 1 [1].

Furthermore, since $BR^n_{\tau_t} \to BR^n$ as $\tau_t \to 0$, then we have $BR^n_{\tau_t} \in BR^n_{\epsilon_t}$ as $\epsilon_t \to 0$. Hence, UDM-FP with decreasing smoothing parameters results almost surely in a GWFP as $t \to \infty$, and thus converges to the NE on the two-player zero-sum games and potential games [2].

$\square$

### A.3.4 Proof of Proposition 4

**Proposition 4** (EGS Enlargement). *Adding a new (meta-)strategy $S_{\boldsymbol{\theta}}$ via Eq.(8) enlarges EGS. Formally, we have:*

$$\text{EGS}(\mathbb{S}^n) \subseteq \text{EGS}(\mathbb{S} \cup S_{\boldsymbol{\theta}}).$$

*Proof.* Since UDM increases the number of the eigenvalues of $\mathcal{L}^n_K$, the new row in the meta-game $\mathcal{M}$ corresponding to the new strategy $S^n$ must be linearly independent to the other rows, and thus it cannot be a convex combination of the other rows. As a result, the EGS is enlarged.

$\square$

### A.3.5 Proof of Proposition 5

**Lemma 2.** *If at any point the population of UDM $\alpha$-PSRO contains a member of an SSCC of the game, then UDM $\alpha$-PSRO will converge a sub-cycle of that SSCC.*

*Proof.* Suppose that a member of one of the underlying game's SSCCs appears in the UDM $\alpha$-PSRO population. This member will induce its own meta-SSCC in the meta-game's response graph. At least one of the members of the underlying game's corresponding SSCC will thus always have positive probability under the $\alpha$-Rank distribution for the meta-game, and the Oracle for this meta-SSCC will always return a member of the underlying game's SSCC. If the Oracle returns a member of the underlying SSCC already in the population, we claim that the corresponding meta-SSCC

already contains a cycle of the underlying SSCC. To see this, note that if the meta-SSCC does not contain a cycle, it must be a singleton. This singleton is either equal or not equal to the full SSCC of the underlying game. In the later case, the Oracle will return a new strategy from the underlying SSCC, contradicting our assumption that it has terminated.

$\square$

**Proposition 5** (Convergence of UDM $\alpha$-PSRO). *In two-player symmetric NFGs, UDM $\alpha$-PSRO converges to the sub-cycle of the unique SSCC.*

*Proof.* From the lemma 2, we only need to prove that there exists a member of one of the underlying game's SSCCs appears in the UDM $\alpha$-PSRO population before it has terminated.

First, the uniqueness of the SSCC follows from the fact that in the two-player symmetric NFGs, the response graph is fully-connected. Suppose at termination of UDM $\alpha$-PSRO, the UDM $\alpha$-PSRO population contains no strategy within the SSCC, and let $S$ be a strategy in the SSCC. We claim that $S$ attains a higher value of the quality term than any strategy in the UDM $\alpha$-PSRO population, which contradicts the fact that UDM $\alpha$-PSRO has terminated. From the definition of SSCC, we know that $S$ has a higher value of quality term than any strategy $S^{'}$ outside the SSCC, and in particular for all $S_i \in \mathbb{S}^1$, and thus the quality term for $S$ is $\exp(1)$. In contrast, for any $S_i \in \mathbb{S}^1$, the quality term for $S_i$ is upper-bounded by $\exp(1 - \boldsymbol{\pi}_i)$. If $\boldsymbol{\pi}_i > 0$, then the quality term of $S_i$ is lower than $S$. If $\boldsymbol{\pi}_i = 0$, then the quality term of $S_i$ is $\exp(0)$. Hence, any strategy in the population has a $< \exp(1)$ quality term, and thus UDM $\alpha$-PSRO cannot terminate before it has encountered an SSCC member. Therefore, we complete the proof.

$\square$

### A.4 Experiment Details

#### A.4.1 Hyper-Parameter Settings

Table 2: Hyper-Parameter Settings for AlphaStar and Blotto.

| Settings | Value | Description |
|---|---|---|
| The Oracle Function | UDM Best Response | Function of Getting Oracles |
| Learning Rate | 0.5 | Learning Rate for Agents |
| Improvement Threshold | 0.03 | Convergence Criteria |
| Meta-Policy Solver | FictitiousPlay | Solve The NE-Policy |
| Meta-Policy Solver Iterations | 1000 | Iterations for Meta-Policy Solver |
| Threads in Pipeline | 2 | Learners in Pipeline PSRO |
| Iterations | 200 | Training Iterations |
| Random Seeds | 5 | Random Seeds of Trials |
| UDM Weighting | 0.15 | Weight of UDM in UDM Best Response |

Table 3: Hyper-Parameter Settings for Non-Transitive Mixture Model.

| Settings | Value | Description |
|---|---|---|
| The Oracle Function | Gradient Ascent | Function of Getting Oracles |
| Optimizer | Adam | Gradient Ascent Optimizer |
| Learning Rate | 0.1 | Learning Rate for Optimizer |
| Betas | (0.9, 0.99) | Betas Parameter for Optimizer |
| $\boldsymbol{\pi}^n = \{\boldsymbol{\pi}^n_k\}_k$ | $\exp\dfrac{(-(x_n - \mu_k)^\mathsf{T}\Sigma(x_n - \mu_k)}{2})$ | Meta-Policy |
| $\Sigma$ | $1/2\mathbf{I}$ | Covariance Matrix for Gaussians |
| $\mu_1$ | (2.871, -0.025) | Position of The First Gaussian |
| $\mu_2$ | (1.8105, 2.2298) | Position of The Second Gaussian |
| $\mu_3$ | (1.8105, -2.2298) | Position of The Third Gaussian |
| $\mu_4$ | (-0.61450, 2.8058) | Position of The Fourth Gaussian |
| $\mu_5$ | (-0.61450, -2.8058) | Position of The Fifth Gaussian |
| $\mu_6$ | (-2.5768, 1.2690) | Position of The Sixth Gaussian |
| $\mu_7$ | (-2.5768, -1.2690) | Position of The Seventh Gaussian |
| Meta-Policy Solver | Fictitious Play | Solve the NE-Policy |
| Meta-Policy Solver Iterations | 1000 | Iterations for Meta-Policy Solver |
| Iterations | 50 | Training Iterations |
| UDM Weight at Iteration $t$ | $\dfrac{0.7}{1 + \exp(-0.25(t - 25))}$ | Weight of UDM in Best Response |
| Threads in Pipeline | 4 | Learners in Pipeline PSRO |
| Random Seeds | 10 | Random Seeds of Trials |

Table 4: Hyper-Parameter Settings for Additional Experiments in Appendix A.4.3.

| Settings | Value | Description |
|---|---|---|
| The Oracle Function | UDM Best Response | Function of Getting Oracles |
| Learning Rate | 0.5 | Learning Rate for Agents |
| Improvement Threshold | 0.03 | Convergence Criteria |
| Meta-Policy Solver | FictitiousPlay | Solve The NE-Policy |
| Meta-Policy Solver Iterations | 1000 | Iterations for Meta-Policy Solver |
| Threads in Pipeline | 2 | Learners in Pipeline PSRO |
| Iterations | 200 | Training Iterations |
| Random Seeds | 5 | Random Seeds of Trials |
| UDM Weighting | 0.15 | Weight of UDM in UDM Best Response |

### A.4.2 Pseudo Codes

---

**Algorithm 1** UDM Gradient Ascent Oracle

---

**Input:** Player population $\mathbb{S}_t = \prod_{n \in [N]} \mathbb{S}_t^n$ with $S_t^n \in \mathbb{S}_t^n$ parameterised by $\theta_{S_t^n}$,
    Meta-policies $\boldsymbol{\pi}_t = \prod_{n \in [N]} \boldsymbol{\pi}_t^n$,
    Number of training updates $N_{\text{train}}$,
    Diversity probability $\lambda$.
**Output:** $S^{\text{train}}$.
1: Randomly initialise a new $S^{\text{train}}$;
2: **for** $j = 1, \cdots, N_{\text{train}}$ **do**
3:     Compute payoff $p_j$ of $S^{\text{train}}$;
4:     Compute meta-payoff $\mathcal{M}_j = \mathcal{M}(\mathbb{S}_t^n \cup \{S^{\text{train}}\})$;
5:     Compute UDM $d_j = \text{Tr}(f(\mathcal{M}_j \mathcal{M}_j^{\mathsf{T}}))$;
6:     Compute loss $l_j = -(1 - \lambda)p_j - \lambda d_j$;
7:     Update $\theta_{S^{\text{train}}}$ to minimise $l_j$ using a gradient based optimization method;
8: **end for**
9: **return** $S^{\text{train}}$

---

---

**Algorithm 2** UDM Best Response Oracle

---

**Input:** Player population $\mathbb{S}_t = \prod_{n \in [N]} \mathbb{S}_t^n$ with $S_t^n \in \mathbb{S}_t^n$ parametrised by $\theta_{S_t^n}$,
    Meta-policies $\boldsymbol{\pi}_t = \prod_{n \in [N]} \boldsymbol{\pi}_t^n$,
    Learning rate $\mu$,
    Diversity probability $\lambda$.
**Output:** $S_t^n$.
1: Compute $\text{BR}_{\text{quality}}^n = \text{BR}^n(\boldsymbol{\pi}_t^{-n})$;
2: **for** each pure strategy $P_j$ **do**
3:     Update meta-payoff $\mathcal{M}_j = \mathcal{M}(\mathbb{S}_t^n \cup \{P_j\})$;
4: **end for**
5: Compute $\text{BR}_{\text{UDM}}^n = \arg \max_{P_j} \text{UDM}(\mathbb{S}_t^n \cup \{P_j\})$;
6: Choose $\text{BR}^n = \text{BR}_{\text{UDM}}^n$ with probability $\lambda$ else $\text{BR}^n = \text{BR}_{\text{quality}}^n$;
7: Update $\theta_{S_t^n} = \mu \theta_{S_t^n} + (1 - \mu)\theta_{\text{BR}^n}$;
8: **return** $S_t^n$

---

### A.4.3 Additional Experiments

#### 1. UDM-FP and UDM $\alpha$-PSRO

Here we validate UDM-FP and UDM $\alpha$-PSRO on the normal-form games. The baselines are FP and $\alpha$-PSRO, respectively.

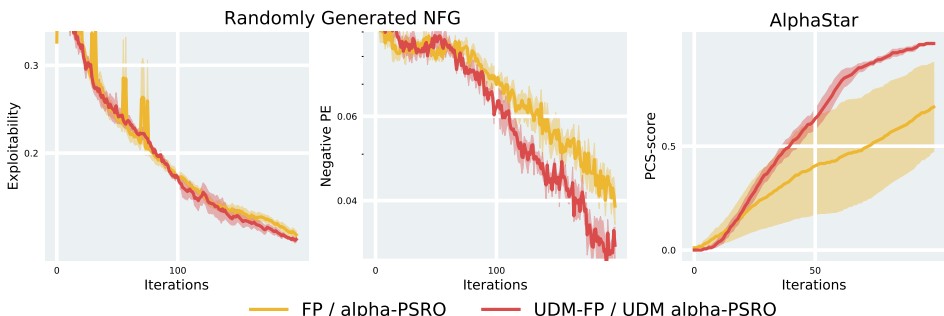

Figure 1: Exploitability & Negative PE / PCS-score vs. Iterations in NFGs.

**UDM-FP.** In the first two subgraphs in Figure 1, we validate UDM-FP on the randomly generated NFGs, and the performances of UDM-FP are better in terms of exploitability and PE. The results also prove the correctness of Proposition 3 that UDM-FP can converge to the NE on two-player zero-sum NFGs.

**UDM $\alpha$-PSRO** Notice that the solution concept of (UDM-)$\alpha$-PSRO is $\alpha$-Rank. PCS-score [1] is a better metric to assess the quality of the population than exploitability that measures the "distance" to a NE. Concretely, PSC-score computes the proportion of current strategies in the empirical game that also belongs to the full games SSCCs. The result in the last subgraph in Figure 1 shows that UDM $\alpha$-PSRO can jump out the current strategic cycles during training and achieve a higher PCS-score than the baseline.

#### 2. Extensive-Form Games

Here we provide two additional experiments consisting of a chess game and a poker game to validate our method. The settings of UDM-PSRO are consistent with the one we introduce in Section 5, and the experimental settings can be found in Appendix A.4.

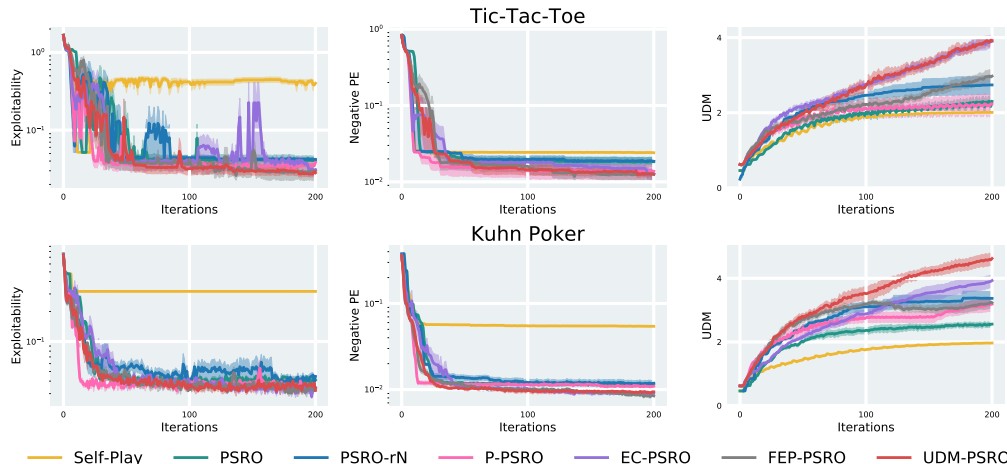

Figure 2: Tic-Tac-Toe and Kuhn Poker: Exploitability & Negative PE & UDM vs. Iterations.

**Tic-Tac-Toe.**  Tic-tac-toe is a type of chess game for two players. The board consists of $3 \times 3$ small squares, which are crossed by two vertical and two horizontal lines. Each player takes turns drawing a cross or a circle in a square, and the player wins if it places three equal pieces in a row, either vertically, horizontally or diagonally. The results in the upper row of Figure 2 show that our method achieves the best expolitability and PE.

**Kuhn Poker.**  Kuhn poker is a zero-sum two-player imperfect-information poker game. In Kuhn poker, the deck includes only three playing cards, for example a King, Queen, and Jack. One card is dealt to each player, which may place bets similarly to a standard poker. If both players bet or both players pass, the player with the higher card wins, otherwise, the betting player wins. We report the performances of different algorithms in this game in the below row of Figure 2, and the results show that our method is competitive with the diversity-aware baselines, and it performs better than other baselines in terms of expolitability and PE.

### 3. Additional Experiments

Here we provide two additional experiments. One is to show that UDM-PSRO can achieve a better performance by considering RD and BD at the same time; the other one on AlphaGO is used to supplement the real-world meta-games in Section 5.

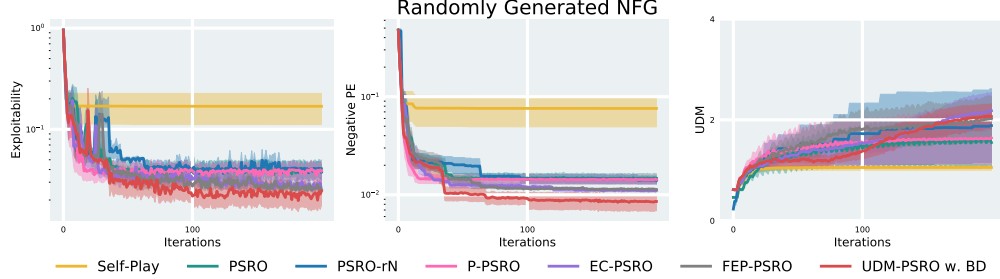

Figure 3: UDM-FP with BD & RD: Exploitability & Negative PE & UDM vs. Iterations.

**UDM-PSRO with RD & BD**  In Figure 3, we report the performances of different algorithms over a normal-form game with 500 meta-strategies. The performance of UDM-PSRO incorporating RD and BD simultaneously is significantly better than the baselines in terms of exploitability and PE.

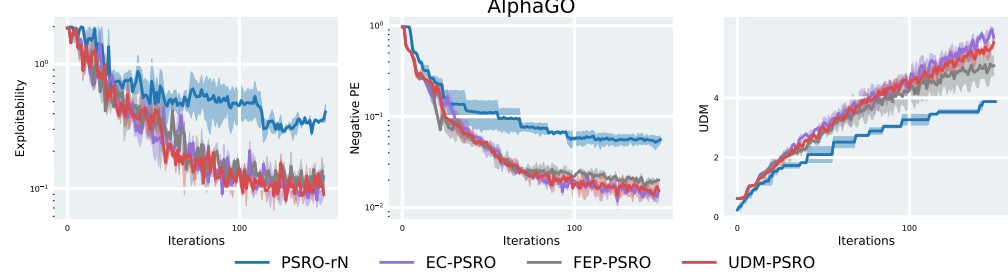

Figure 4: UDM-PSRO on AlphaGO: Exploitability & Negative PE & UDM vs. Iterations.

**AlphaGO.**  We investigate the diversity-aware algorithms on AlphaGO with 1679 meta-strategies. The results in Figure 4 show that UDM-PSRO is competitive with most baselines, while PSRO-rN fails to recover the diverse strategies and is easily exploited.

### A.5 Discussions

**1) Selection of $f(x)$ and $K\langle x, y\rangle$.** We firstly discuss about the selection of $f(x)$ and $K\langle x, y\rangle$. As for the function $f(x)$, the principle of choosing $f(x)$ is that the function should be bounded, monotonically increasing, and $f(0) = 0$ (refer to Section 3.1 for more explanations). There are lots of functions that satisfy these properties, e.g., $f(x) = \frac{g(x)}{\gamma+g(x)} - \frac{g(0)}{\gamma+g(0)}$, where $\gamma > 0$ is a constant, $g(x)$ is a monotonically increasing function and $g(0) \geq 0$. In our paper, we choose $g(x) = \exp(x)$ since $f(x) = \frac{1}{1+\gamma\exp(-x)} - \frac{1}{1+\gamma}, \gamma \in (0, 1]$ has a sufficiently large convergence region $R = (0, \infty)$. The results in Figure 5 show that $\gamma = 1$ is the best.

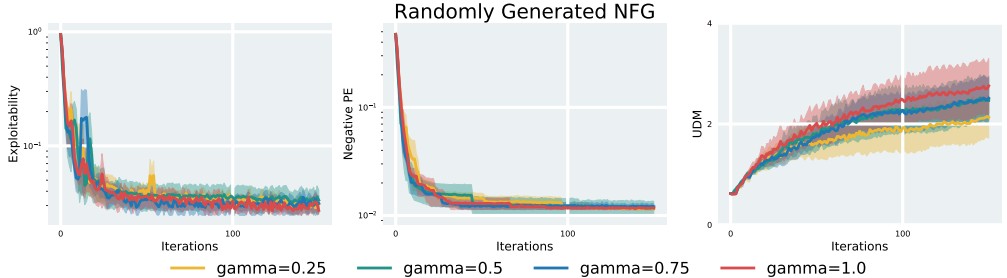

Figure 5: Ablation Study of $\gamma$: Exploitability & Negative PE & UDM vs. Iterations.

As for the diversity kernel, we can choose some simple but effective kernel functions such as the linear kernel, polynomial kernel and Gaussian kernel. Since the dimension of the feature vector (i.e. $\mathcal{M}_i$) in our experiments is large, the computational burden of Gaussian kernel would be higher than the others. We finally use $K\langle x, y\rangle = (\langle x, y\rangle + 1)^3$ due to its best performances in the ablation study shown in Figure 6.

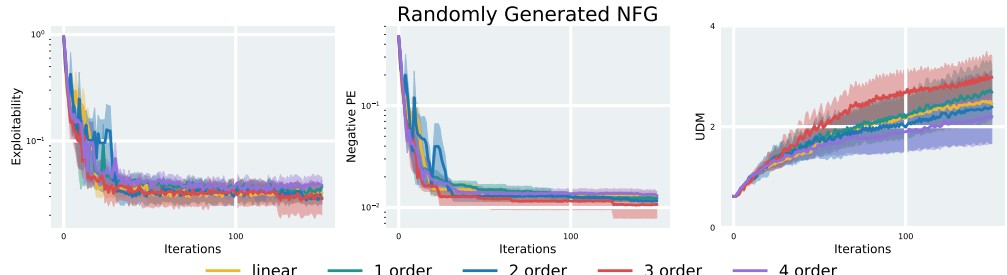

Figure 6: Ablation Study of Kernels: Exploitability & Negative PE & UDM vs. Iterations.

**2) Extension to More General Games.** We here discuss whether UDM can be extended to n-player, general-sum, or non-symmetric games or not. In theory, UDM can still work in these games. For each player $n$, UDM measures the diversity of a population through the diversity kernel $[K(\phi_i, \phi_j)]_{i,j}$, which is determined by the strategy features $\{\phi_i\}_i$ of the population. Thus, to show that UDM can still work in these games, it suffices to show that the strategy features $\{\phi_i\}_i$ are independent of the types of games. Concretely, we can choose $\phi_i = \mathcal{M}_{[i,:]}^{(n)}$, where $\mathcal{M}_{i,j}^{(n)} :=$ $\sum_{S^n}\sum_{S^{-n}} \pi_i^{(n)}(S^n)\cdot g^n(S^n, S^{-n})\cdot\pi_j^{(-n)}(S^{-n})$ is the utility of the $i$-th strategy $\pi_i^{(n)}$ of the player $n$ against the $j$-th joint strategy $\pi_j^{(-n)}$ of the players $-n$. However, since the length of $S^{-n} :=$ $(S^1, \cdots, S^{n-1}, S^{n+1}, \cdots, S^N)$ increases with the number of the players, the computational cost of UDM would be expensive. Investigating how to reduce the computational cost in n-player, general-sum, or non-symmetric games can be an important future work.