# OpenReview forum: "A Unified Diversity Measure for Multiagent Reinforcement Learning"
_NeurIPS.cc/2022/Conference — NeurIPS 2022 Accept_

### Official Review · Reviewer_67AT · 2022-07-10

**Rating:** 6
**Confidence:** 4
**Soundness:** 3 good
**Presentation:** 4 excellent
**Contribution:** 3 good

**Summary:**

This paper presents a unified diversity measure (UDM) for MARL learning. By choosing different diversity kernels and the function $f$, UDM can recover different existing diversity metrics. The authors establish some convergence properties under UDM and then conduct several experiments (3 in the main text and 2 additional in the appendix) including both transitive and non-transitive games. The experimental results show that UDM outperforms baselines without explicit diversity objectives and is comparable to baselines with diversity objectives in terms of exploitability and population effectivity.

**Questions:**

The questions are mainly concerning the instantiation of UDM.

(1) Could you elaborate why the kernel function $K\langle x,y \rangle=(\langle x,y\rangle+1)^3$ and $f(x)=\frac{1}{1+\exp(-x)}-\frac{1}{2}$ is chosen? Or, is there any principle for choosing these two functions in UDM?

(2) Will UDM-FP and UDM-$\alpha$-PSRO perform better (in terms of expl and PE) than the baselines that used FP and $\alpha$-PSRO respectively?

(3) This work seems to focus on RD, while UDM can be readily applied to BD. FEP-PSRO unified BD and RD, and the experiments showed it is helpful. Can UDM incorporate RD and BD simultaneously and achieve better performance than only considering one of RD and BD?

(4) The convergence of UDM-FP is provided by showing it is a GWFP process. Is it possible to show a faster convergence speed with this diversity term?

**Limitations:**

I favor the motivation of this paper, and the proposed method can provide a better perspective on the population diversity in MARL (especially in open-ended learning for two-player zero-sum games). It would be very helpful to provide an ablation study so that future researchers can know how to instantiate UDM.

**Strengths And Weaknesses:**

**Strengths**:

This study is closely related to the learning for MARL. With a more diverse population, we are expected to achieve a faster learning speed to the target objective (e.g., lower exploitability, higher population effectivity). However, existing diversity metrics may be motivated by different practical observations and no well-defined unified diversity metric for the learning of multi-agent systems. UDM serves this purpose well by unifying existing diversity metrics into a single function, which may help to design new learning algorithms.

I think this paper is well-written and easy to follow. Based on a diversity kernel matrix, the authors show how UDM correlates to and differs from existing diversities. Some benefits of UDM are also revealed -- UDM can not only recover ED and PD separately but also tackle some of the notorious problems (ignoring weak but useful strategies and cannot distinguish redundant strategies).

Besides, UDM also seems technical sound where the methods and proofs are intuitive.


**Weaknesses**:

Current implementations (the choice of the kernel function and $f$) and experiment results are less informative. In the experiment part, the authors focus on $K\langle x,y \rangle=(\langle x,y\rangle+1)^3$ and $f(x)=\frac{1}{1+\exp(-x)}-\frac{1}{2}$ and only RD metrics. UDM generalizes RD and BD, while only a small part of the generation is shown in the current version. From the experiments, in terms of exploitability (expl) and PE, the differences among UDM, EC-PSRO, and FEP-PSRO are indistinguishable in AlphaStar888 and Blotto. I noticed that the experiments on two extensive-form games also demonstrated the above results, and the expl and PE of FEP-PSRO seems to be slightly better. The experiments for UDM-FP and UDM-$\alpha$-PSRO are missing. More insights (empirical or theoretical) from other instantiations of UDM are expected, which can be achieved by an ablation study.

---

> ### Author Response · Authors · 2022-08-02
> **Response to Reviewer 67AT**
>
> We thank the reviewer for all these valuable comments. We provide point-by-point responses below.
>
> **Q1: "Could you elaborate why the kernel function $K⟨x,y⟩=(⟨x,y⟩+1)^3$ and $f(x)=\frac{1}{1+exp⁡(−x)}−\frac{1}{2}$ is chosen? Or, is there any principle for choosing these two functions in UDM?"**
>
> **A1:** As for the function $f(x)$, the principle of choosing $f(x)$ is that the function should be bounded, monotonically increasing, and $f(0)=0$ (refer to Section 3.1 for more explanations). There are lots of functions that satisfy these properties, e.g., $f(x)=\frac{g(x)}{\gamma+g(x)}-\frac{g(0)}{\gamma+g(0)}$, where $\gamma>0$ is a constant, $g(x)$ is a monotonically increasing function and $g(0)\ge0$. In our paper, we choose $g(x)=\exp(x)$ since $f(x)=\frac{1}{1+\gamma\exp(-x)}-\frac{1}{1+\gamma}, \gamma\in(0,1]$ has a sufficiently large convergence region $R=(0,\infty)$. We have added an ablation study on $\gamma$ and it shows that $\gamma=1$ is the best, as shown below.
>
> | $\gamma$ | exploitabiliaty | negative PE |
> | :------: | :-------------: | :---------: |
> |  $0.25$  |      0.031      |    0.012    |
> |  $0.50$  |      0.033      |    0.012    |
> |  $0.75$  |      0.031      |    0.012    |
> |  $1.00$  |    **0.025**    |  **0.010**  |
>
> As for the diversity kernel, we can choose some simple but effective kernel functions such as the linear kernel, polynomial kernel and Gaussian kernel. Since the dimension of the feature vector (i.e. $\mathcal{M}_{[i,:]}$) in our experiments is large, the computational burden of Gaussian kernel would be higher than the others. We finally use $K\langle{x,y}\rangle=(\langle{x,y}\rangle+1)^{3}$ due to its best performance in the ablation study as shown below.
>
> |      kernel function      | exploitability | negative PE |
> | :-----------------------: | :------------: | :---------: |
> |       linear kernel       |     0.032      |    0.013    |
> | 1-order polynomial kernel |     0.037      |    0.012    |
> | 2-order polynomial kernel |     0.029      |    0.012    |
> | 3-order polynomial kernel |   **0.025**    |  **0.010**  |
> | 4-order polynomial kernel |     0.038      |    0.013    |
>
>
> All the above experiments have been added in Appendix A.5 (page 16, lines 230-241, Appendix).
>
> **Q2: "Will UDM-FP and UDM-α-PSRO perform better (in terms of expl and PE) than the baselines that used FP and α-PSRO respectively?"**
>
> **A2:** We have made additional experiments of UDM-FP and UDM $\alpha$-PSRO, and the results in Appendix A.4.3 (page 14, lines 189-201, Appendix) show that UDM $\alpha$-PSRO and UDM-FP perform better than $\alpha$-PSRO and FP respectively. Since the solution concept of (UDM-)$\alpha$-PSRO is $\alpha$-Rank, PCS-score is adopted as a metric to assess the quality of the population insteading of exploitability, as argued in [1].
>
> * |      method       | PCS-score |
>     | :---------------: | :-------: |
>     |   $\alpha$-PSRO   |   0.68    |
>     | UDM $\alpha$-PSRO | **0.99**  |
>
> * | method | exploitability | negative PE |
>     | :----: | :------------: | :---------: |
>     |   FP   |      0.14      |    0.04     |
>     | UDM-FP |    **0.13**    |  **0.03**   |
>
> **Q3: "Can UDM incorporate RD and BD simultaneously and achieve better performance than only considering one of RD and BD?"**
>
> **A3:** Yes, UDM-PSRO can achieve a better performance by considering RD and BD at the same time as shown in the following table.
>
> |      method       | exploitability | negative PE |
> | :---------------: | :------------: | :---------: |
> |     Self-Play     |      0.17      |    0.076    |
> |       PSRO        |      0.04      |    0.015    |
> |      PSRO-rN      |      0.04      |    0.014    |
> |      P-PSRO       |      0.04      |    0.014    |
> |      EC-PSRO      |      0.03      |    0.011    |
> |     FEP-PSRO      |      0.03      |    0.011    |
> | UDM-PSRO w. RD&BD |    **0.02**    |  **0.008**  |
>
> We have added these results in Appendix A.4.3 (page 15, lines 222-224, Appendix).
>
> **Q4: "The convergence of UDM-FP is provided by showing it is a GWFP process. Is it possible to show a faster convergence speed with this diversity term?"**
>
> **A4:** Intuitively, since the diversity term encourages UDM-FP to explore the strategy space, UDM-FP could find the best strategy faster and thus converge faster, which is also validated by the empirical results in Appendix A.4.3 (page 14, lines 192-195, Appendix). However, a strict theoretical proof is not straightforward and we leave it to our future work.
>
> ---
>
> ref.
>
> [1] Muller et al., A generalized training approach for multiagent learning, ICLR 2019.
>
> ---

---

> > ### Comment · Reviewer_67AT · 2022-08-04
> > **Response to Authors**
> >
> > Thanks for the response and the additional experiments!

---

### Official Review · Reviewer_Bqe8 · 2022-07-11

**Rating:** 5
**Confidence:** 3
**Soundness:** 3 good
**Presentation:** 2 fair
**Contribution:** 2 fair

**Summary:**

This paper offers a unified diversity measure for multi-agent reinforcement learning. The authors first show the existing diversity measure and then provide Unified Diversity Measure (UDM) from a geometric perspective to unify all existing diversity measures. After showing the relationship between UDM and existing diversity measures, the authors provide two algorithms, UDM Fictitious Play and UDM PSRO, to provide diversity policies. Experiments on AlphaStar and Blotto show the performance of the proposed method over baselines.

**Questions:**

- Add the background of the geometric intuition.
- Polish the notation and the writing of this paper for more general audiences, not only for the game theory community but also the reinforcement learning/machine learning community.
- How to generate strategy feature and diversity kernel when applying UDM in the new games.
- What if the number of agents is not two?
- Line 293, missing the experiments of AlphaGO.
- If a game has NE, why do we need to explore the diversity, especially when we can get the whole payoff matrix.
- I am curious about the experiments on AlphaStar. AlphaStar is trained for about 14 days with 16 TPUs for each agent. How to get meta-payoffs for 888 RL meta-strategies?

- Minor
    - Line 203 the meaning of Hadamard product of a vector and a matrix
    - The meaning of $|| \cdot ||_{F}$ above line 206
    - What is $m^*_i$ in line 207
    - Line 186 “similar with” -> “similar to”


**Limitations:**

I didn't find the description of the limitation of the proposed methods. And the authors didn't discuss any potential negative societal impacts. The author can describe the limitation, for example, is the proposed method limited to the two-player zero-sum game.

**Strengths And Weaknesses:**

- Strengths
    - The unified framework reveals the similarity among various diversity measures, and provides a new view of the diversity of policies.
    - The overall writing flow is good

- Weaknesses
    - Lack of the background of the geometric intuition.
    - The notations are too much and confusing, which are unfriendly to the audiences outside the game theory community.

---

> ### Author Response · Authors · 2022-08-02
> **Response to Reviewer Bqe8**
>
> We thank the reviewer for all these valuable comments. We provide point-by-point responses below.
>
> **Q1: "Add the background of the geometric intuition."**
>
> **A1:** We have added the background of the geometric intuition in the revised version (page 4, lines 163-166, in paper).
>
> **Q2: "Polish the notation and the writing of this paper for more general audiences, not only for the game theory community but also the reinforcement learning/machine learning community."**
>
> **A2:** Thanks for this comment. The notations and writing of this paper generally follow previous work [1]. According to your advice, we have added a table of notations in Appendix A.1.1 (page 2, line 24, Appendix) to further impove the readability.
>
> **Q3: "How to generate strategy feature and diversity kernel when applying UDM in the new games."**
>
> **A3:** As for the strategy feature,  we can choose $\phi_{i}=\mathcal{M}_{[i,:]}$ if we focus on RD (response diversity),
>
> or $\phi_{i}=\\{\pi_{i}(\cdot|s)\\}_{s}$ for BD (behavioral diversity).
>
> As for the diversity kernel, we can choose some simple but effective kernel functions such as the linear kernel, polynomial kernel and Gaussian kernel. Since the dimension of the feature vector (i.e. $\mathcal{M}_{[i,:]}$) in our experiments is large, the computational burden of Gaussian kernel would be much higher than the others. We finally choose $K\langle{x,y}\rangle=(\langle{x,y}\rangle+1)^{3}$ due to its best performance as shown below.
>
> |      kernel function      | exploitability | negative PE |
> | :-: | :-: | :-: |
> |       linear kernel       |     0.032      |    0.013    |
> | 1-order polynomial kernel |     0.037      |    0.012    |
> | 2-order polynomial kernel |     0.029      |    0.012    |
> | 3-order polynomial kernel |   **0.025**    |  **0.010**  |
> | 4-order polynomial kernel |     0.038      |    0.013    |
>
> The above results have been added in Appendix A.5 (page 16, lines 230-241, Appendix).
>
> **Q4: "What if the number of agents is not two?"**
>
> **A4:** Theoretically, UDM can still work in n-player games. For each player $n$, UDM measures the diversity of a population through the diversity kernel $[K(\phi_{i},\phi_{j})]$, which is determined by the strategy features $\\{\phi_{i}\\}$ of the population. Thus, to show that UDM can still work in multi-player games, it suffices to show that the strategy features $\\{\phi_{i}\\}$ are independent of the types of games. Concretely, we can choose $\phi_{i}=\mathcal{M}_{[i,:]}^{(n)}$, where
>
> $\mathcal{M}_{i,j}^{(n)}$
>
> $:=\sum_{S^{n}}\sum_{S^{-n}}\pi_{i}^{(n)}(S^{n})\cdot g^{n}(S^{n},S^{-n})\cdot\pi_{j}^{(-n)}(S^{-n})$
>
> is the utility of the $i$-th policy $\pi_{i}^{(n)}$ of the player $n$ against the $j$-th joint policy $\pi_{j}^{(-n)}$ of the players $-n$. However, since the length of joint strategy $S^{-n}:=(S^{1},\cdots,S^{n-1},S^{n+1},\cdots,S^{N})$ increases with the number of the players, the computational cost of UDM would be expensive. Investigating how to reduce the computational cost when extending UDM to n-player games can be an important future work.
>
> We have added the above explanations in Appendix A.5 (page 16, lines 242-252, Appendix).
>
> **Q5: "Line 293, missing the experiments of AlphaGO."**
>
> **A5:** In AlphaGO, the following numerical results show that our method performs better than the diversity-aware baselines.
>
> |  method  | exploitability | negative PE |
> | :-: | :-: | :-: |
> | PSRO-rN  |      0.41      |    0.06     |
> | EC-PSRO  |      0.13      |    0.02     |
> | FEP-PSRO |    **0.09**    |    0.02     |
> | UDM-PSRO |    **0.09**    |  **0.01**   |
>
> The above results have been added in Appendix A.4.3 (page 15, lines 225-227, Appendix).

---

> > ### Author Response · Authors · 2022-08-02
> > **Response to Reviewer Bqe8**
> >
> > **Q6: "If a game has NE, why do we need to explore the diversity, especially when we can get the whole payoff matrix."**
> >
> > **A6:** In theory, we can compute its NE if we have the whole payoff matrix of a game. However, it is computationally expensive to search for the NE directly when the game size is large since no polynomial-time solution is available even in 2-player cases [2]. An iterative method, such as PSRO, PSRO-rN, etc., is therefore a better solution with lower computational cost, but at the same time, might encounter the diversity issues. As discussed in the RPS-X game (Appendix A.2.1), PSRO-rN fails to find the best strategy X (i.e., the NE), but promoting the strategy diversity in the iterative process can tackle this problem properly.
> >
> > **Q7: "I am curious about the experiments on AlphaStar. AlphaStar is trained for about 14 days with 16 TPUs for each agent. How to get meta-payoffs for 888 RL meta-strategies?"**
> >
> > **A7:** The work [4] derives the meta-payoff matrices of some complex real-world games including AlphaStar to analyze the non-transitive properties in these games. These meta-payoff matrices have been used to validate the diversity-aware algorithms in subsequent work like [1] [5]. We also use this meta-payoff matrix to validate our methods.
> >
> > **Q8: About the Minor**
> >
> > 1) Line 203 the meaning of Hadamard product of a vector and a matrix
> >
> > 2) The meaning of $\Vert\cdot\Vert_F$ above line 206
> >
> > 3) What is $m_i$* in line 207
> >
> > 4) Line 186 “similar with” -> “similar to”
> >
> > **A8:**
> >
> > 1) Hadamard product $\odot$ is defined as $A\odot B := [a_{ij} b_{ij}]$, where $A=[a_{ij}], B = [b_{ij}]$. (page 6, line 206, in paper)
> >
> > 2) $\Vert\cdot\Vert_F$ usually refers to the Frobenius norm, which is defined as $\Vert{A}\Vert_F:=\sqrt{\sum_{i,j}a_{ij}^2}=\sqrt{\textrm{Tr}(A^{\mathsf{T}}A)}$, where $A=[a_{ij}]$. (page 6, line 207, in paper)
> >
> > 3) $m_i$* is the $i$-th row of $\mathcal{M}^*$. (page 6, line 209, in paper)
> >
> > 4) We have proofreaded the paper again to elliminate any potential typos. (page 5, line 188, in paper)
> >
> >
> > **Q9: "I didn't find the description of the limitation of the proposed methods. And the authors didn't discuss any potential negative societal impacts. The author can describe the limitation, for example, is the proposed method limited to the two-player zero-sum game."**
> >
> > **A9:** Thank for your comment. We have added the limitations of our method in the Discussions in Appendix A.5 (page 16, lines 242-252, Appendix).
> >
> > ----
> >
> > ref.
> >
> > [1] Nieves et al., Modelling behavioural diversity for learning in open-ended games, ICML 2021.
> >
> > [2] Chen et al., Settling the complexity of computing two-player nash equilibria, JACM 2009.
> >
> > [3] McMahan et al., planning in the presence of cost functions controlled by an adversary, ICML 2003.
> >
> > [4] Czarneck et al., Real world games look like spinning tops, NeurIPS 2020.
> >
> > [5] Liu et al., Towards unifying behavioral and response diversity for open-ended learning in zero-sum games, NeurIPS 2021.

---

> > > ### Comment · Reviewer_Bqe8 · 2022-08-09
> > > **Re: Response to Reviewer Bqe8**
> > >
> > > Thanks for your response and most of my concerns are addressed. I would raise my evaluation.

---

### Official Review · Reviewer_6GhH · 2022-07-11

**Rating:** 7
**Confidence:** 3
**Soundness:** 3 good
**Presentation:** 3 good
**Contribution:** 3 good

**Summary:**

This paper proposes a unifying diversity measure of three diversity measures (ED, PD, and EC), and uses the unification to explain properties of the existing metrics. UDM-PSRO is proposed which uses the new diversity metric as an oracle. It is compared to other baselines on some simple normal-form games.

**Questions:**

Q1: Can this diversity measure be easily extended to n-player, general-sum, or non-symmetric games?

**Limitations:**

The usual limits on symmetric tow-player zero-sum should be states again in the conclusion please.

**Strengths And Weaknesses:**

Strengths: This paper is well-written and clear to follow. The maths and proofs seem sound to me. The key idea is interesting and important research.

Weaknesses: Exploitability experiments show similar performance to other pre-existing methods. Exploitability of extensive-form games was not evaluated.

I think this is a strong paper. More thorough evaluation on (real) extensive form games would make it stronger.

Comments:

Experiments Section (Figure 1): I believe all the experiments are made on normal-form games? Therefore it might be good to mention/cite double oracle as well as PSRO.

---

> ### Author Response · Authors · 2022-08-02
> **Response to Reviewer 6GhH**
>
> We thank the reviewer for all these valuable comments. We provide point-by-point responses below.
>
> **Q1: "Can this diversity measure be easily extended to n-player, general-sum, or non-symmetric games?"**
>
> **A1**: Theoretically, UDM can still work in n-player, general-sum, or non-symmetric games. For each player $n$, UDM measures the diversity of a population through the diversity kernel $[K(\phi_{i},\phi_{j})]$, which is determined by the strategy features $\\{\phi_{i}\\}$
> of the population. Thus, to show that UDM can still work in these games, it suffices to show that the strategy features $\\{\phi_{i}\\}$ are independent of the types of games. Concretely, we can choose $\phi_{i}=\mathcal{M}_{[i,:]}^{(n)}$, where
>
> $\mathcal{M}_{i,j}^{(n)}$
>
> $:=\sum_{S^{n}}\sum_{S^{-n}}\pi_{i}^{(n)}(S^{n})\cdot g^{n}(S^{n},S^{-n})\cdot\pi_{j}^{(-n)}(S^{-n})$
>
> is the utility of the $i$-th policy $\pi_{i}^{(n)}$ of the player $n$ against the $j$-th joint policy $\pi_{j}^{(-n)}$ of the players $-n$. However, since the length of joint strategy $S^{-n}:=(S^{1},\cdots,S^{n-1},S^{n+1},\cdots,S^{N})$ increases with the number of the players, the computational cost of UDM would be expensive. Investigating how to reduce the computational cost when extending UDM to n-player, general-sum, or non-symmetric games can be an important future work.
>
> The above explanations have been added in Discussions in Appendix A.5 (page 16, lines 242-252, Appendix).
>
> **Q2: "Exploitability of extensive-form games was not evaluated."**
>
> **A2**: Actually, we provided the results of extensive-form games including Kuhn Poker and Tic-Tac-Toe in Apendix A.4.3 (pages 14-15, lines 202-217, Appendix) and the results show that our method achieves the lower exploitability than the non-diversity baselines.
>
> **Q3: "Experiments Section (Figure 1): I believe all the experiments are made on normal-form games? Therefore it might be good to mention/cite double oracle as well as PSRO."**
>
> **A3**: The experiments provided in Section 5 are investigated on normal-form games. As shown in Table 1 (page 3, line 110, in paper), double oracle is an instance of PSRO with $N=2$ and the policy solver $\mathcal{S}$ set to the NE in normal-form games [1] [2]. Therefore, the performance of double oracle is consistent with PSRO's in normal-form games provided $\mathcal{S}=\textrm{NE}$ and $\mathcal{O}=\textrm{BR}(\cdot)$. We have cited double oracle in Table 1 for comparision of the existing main game solvers.
>
> ---
>
> ref.
>
> [1] Lanctot et al., A unified game-theoretic approach to multiagent reinforcement learning, NeurIPS 2017.
>
> [2] Balduzzi et al., Open-ended learning in symmetric zero-sum games, PMLR 2019.
>
> ---

---

> > ### Comment · Reviewer_6GhH · 2022-08-03
> > **Reply**
> >
> > Thank you for the answers.

---

### Meta-Review · Area_Chair_XN8V · 2022-08-24

**Recommendation:** Accept
**Confidence:** Certain

**Metareview:**

This paper provides a unifying framework for promoting diverse behaviors in multi-agent RL. The framework---the unified diversity measure--- is general enough to be able to capture several other recently proposed measures as special cases (associated with specific kernel functions). The paper then provides extensions two MARL algorithms (PSRO and Fictitious-play) which make use of UDM to promote diverse behaviors in MARL and show that they converge asymptotically to relevant equilibria and provide numerical examples.

Reviewers were generally positive on the paper, finding it well written and proposing an interesting idea for promoting diversity in MARL that seemed intuitive.

**Award:**

No

---

### Decision · Program_Chairs · 2022-09-14

Accept